# Development, calibration, and validation of a novel human ventricular myocyte model in health, disease, and drug block

Jakub Tomek[1]*, Alfonso Bueno-Orovio[1], Elisa Passini[1], Xin Zhou[1], Ana Minchole[1], Oliver Britton[1], Chiara Bartolucci[2], Stefano Severi[2], Alvin Shrier[3], Laszlo Virag[4], Andras Varro[4], Blanca Rodriguez[1]*

[1]Department of Computer Science, British Heart Foundation Centre of Research Excellence, University of Oxford, Oxford, United Kingdom; [2]Department of Electrical, Electronic, and Information Engineering "Guglielmo Marconi", University of Bologna, Bologna, Italy; [3]Department of Physiology, McGill University, Montreal, Canada; [4]Department of Pharmacology and Pharmacotherapy, Faculty of Medicine, University of Szeged, Szeged, Hungary

**Abstract** Human-based modelling and simulations are becoming ubiquitous in biomedical science due to their ability to augment experimental and clinical investigations. Cardiac electrophysiology is one of the most advanced areas, with cardiac modelling and simulation being considered for virtual testing of pharmacological therapies and medical devices. Current models present inconsistencies with experimental data, which limit further progress. In this study, we present the design, development, calibration and independent validation of a human-based ventricular model (ToR-ORd) for simulations of electrophysiology and excitation-contraction coupling, from ionic to whole-organ dynamics, including the electrocardiogram. Validation based on substantial multiscale simulations supports the credibility of the ToR-ORd model under healthy and key disease conditions, as well as drug blockade. In addition, the process uncovers new theoretical insights into the biophysical properties of the L-type calcium current, which are critical for sodium and calcium dynamics. These insights enable the reformulation of L-type calcium current, as well as replacement of the hERG current model.

*For correspondence:
jakub.tomek.mff@gmail.com (JT);
Blanca.Rodriguez@cs.ox.ac.uk (BR)

Competing interests: The authors declare that no competing interests exist.

## Introduction

Human-based computer modelling and simulation are a fundamental asset of biomedical research. They augment experimental and clinical research through enabling detailed mechanistic and systematic investigations. Owing to a large body of research across biomedicine, their credibility has expanded beyond academia, with vigorous activity also in regulatory and industrial settings. Thus, human in silico clinical trials are now becoming a central paradigm, for example, in the development of medical therapies (*Pappalardo et al., 2018*). They exploit mature human-based modelling and simulation technology to perform virtual testing of pharmacological therapies or devices.

Human cardiac electrophysiology is one of the most advanced areas in physiological modelling and simulation. Current human models of cardiac electrophysiology include detailed information on the ionic processes underlying the action potential such as the sodium, potassium and calcium ionic currents, exchangers such as the Na/Ca exchanger and pumps such as the Na/K pump. They also include representation of the excitation-contraction coupling system in the sarcoplasmic reticulum, an important modulator of the calcium transient, through the calcium-induced calcium-release mechanisms and the SERCA pump. Several human models have been proposed for ventricular electrophysiology, and amongst them the ORd model (*O'Hara et al., 2011*). Its key strengths are the

**eLife digest** Decades of intensive experimental and clinical research have revealed much about how the human heart works. Though incomplete, this knowledge has been used to construct computer models that represent the activity of this organ as a whole, and of its individual chambers (the atria and ventricles), tissues and cells. Such models have been used to better understand life-threatening irregular heartbeats; they are also beginning to be used to guide decisions about the treatment of patients and the development of new drugs by the pharmaceutical industry.

Yet existing computer models of the electrical activity of the human heart are sometimes inconsistent with experimental data. This problem led Tomek et al. to try to create a new model that was consistent with established biophysical knowledge and experimental data for a wide range of conditions including disease and drug action.

Tomek et al. designed a strategy that explicitly separated the construction and validation of a model that could recreate the electrical activity of the ventricles in a human heart. This model was able to integrate and explain a wide range of properties of both healthy and diseased hearts, including their response to different drugs. The development of the model also uncovered and resolved theoretical inconsistencies that have been present in almost all models of the heart from the last 25 years. Tomek et al. hope that their new human heart model will enable more basic, translational and clinical research into a range of heart diseases and accelerate the development of new therapies.

representation of CaMKII signalling, capability to manifest arrhythmia precursors such as alternans and early afterdepolarisation, and good response to simulated drug block and disease remodelling (*Dutta et al., 2016*; *Dutta et al., 2017a*; *Passini et al., 2016*; *Tomek et al., 2017*). Consequently, ORd was selected by a panel of experts as the model best suited for regulatory purposes (*Dutta et al., 2017a*).

Most of the ORd model development has focused on repolarisation properties such as its response to drug block, repolarisation abnormalities and its rate dependence. However, a more holistic comparison of ORd-based simulations with human ventricular experimental data reveals important inconsistencies. Firstly, the plateau of the action potential (AP) is significantly higher in the ORd model than in experimental data used for ORd model construction (*O'Hara et al., 2011*; *Britton et al., 2017*) and in data from additional studies using human cardiomyocytes (*Coppini et al., 2013*; *Jost et al., 2013*). Secondly, the dynamics of accommodation of the AP duration (APD) to heart rate acceleration, which are known to be modulated by sodium dynamics, show only limited agreement with a comparable experimental dataset (*Franz et al., 1988*; *O'Hara et al., 2011*). Thirdly, we identify that simulations of the sodium current block has an inotropic effect in the ORd model, increasing the amplitude of the calcium transient, in disagreement with its established negatively inotropic effect in experimental/clinical data (encainide, flecainide, and TTX) (*Gottlieb et al., 1990*; *Tucker et al., 1982*; *Legrand et al., 1983*; *Bhattacharyya and Vassalle, 1982*). All those properties, namely AP plateau potential, APD adaptation and response to sodium current block, have strong dependencies on sodium and calcium dynamics. We therefore hypothesise that ionic balances during repolarisation require further research. We specifically focus on an in-depth re-evaluation of the L-type calcium current ($I_{CaL}$) formulation, given its fundamental role in determining the AP, the calcium transient and sodium homeostasis through the Na/Ca exchanger. The second main focus is the re-assessment of the rapid delayed rectifier current ($I_{Kr}$), the dominant repolarisation current in human ventricle, under conditions that reflect experimental data-driven plateau potentials.

Using a development strategy based on strictly separated model calibration and validation, we sought to design, develop, calibrate and validate a novel model of human ventricular electrophysiology and excitation contraction coupling, the ToR-ORd model (for Tomek, Rodriguez – following ORd). Our aim for simulations using the ToR-ORd model is to be able to reproduce all key depolarisation, repolarisation and calcium dynamics properties in healthy ventricular cardiomyocytes, under drug block, and in key diseased conditions such as hyperkalemia (central to acute myocardial ischemia), and hypertrophic cardiomyopathy.

## Materials and methods

### Strategy for construction, calibration and validation of the ToR-ORd model

*Table 1* lists the properties (left column) and key references (right column) of experimental and clinical datasets considered for the calibration (top) and independent validation (bottom) of the ToR-ORd model. This represents a comprehensive list of properties, known to characterize human ventricular electrophysiology under multiple stimulation rates, and also drug action and disease. The recordings in were obtained in human ventricular preparations primarily using measurements with microelectrode recordings, unipolar electrograms, and monophasic APs, therefore avoiding photon scattering effects or potential dye artefacts present in optical mapping experiments. In addition, the ToR-ORd model was calibrated to manifest depolarisation of resting membrane potential in response to an $I_{K1}$ block, based on evidence in a range of studies summarised in *Dhamoon and Jalife (2005)*. The calibration criteria are chosen to be fundamental properties of ionic currents, action potential and single-cell pro-arrhythmic phenomena (described in more detail in Appendix 1-1). The validation criteria include response to rate changes, drug action and disease, to explore the predictive power of the model under clinically-relevant conditions.

We initially performed the evaluation of the ORd model (*O'Hara et al., 2011*) by conducting simulations for each of the calibration criteria in *Table 1*. Further details are described throughout the Materials and methods section and Appendix 1-15.1. Simulations with the existing versions of the ORd model failed to fulfil key criteria such as AP morphology, calcium transient duration, several properties of the L-type calcium current, negative inotropic effect of sodium blockers, or the depolarising effect of $I_{K1}$ block. The results are later demonstrated in Figures 2 and 3, and *Methods: Calibration of $I_{K1}$ block and resting membrane potential*. Secondly, we attempted parameter optimisation using a multiobjective genetic algorithm (*Torres et al., 2012*). However, simulations with the ORd-based models were unable to fulfil key criteria such as AP and Ca morphology, and the effect of sodium and calcium block on calcium transient amplitude and APD, respectively.

We then proceeded to reevaluate the ionic current formulations based on experimental data and biophysical knowledge. Key currents included $I_{CaL}$ and specifically its driving force and activation, as

**Table 1.** Criteria and human-based studies used in ToR-ORd calibration and validation.

**Calibration**

| | |
|---|---|
| Action potential morphology | (*Britton et al., 2017*; *Coppini et al., 2013*; *Jost et al., 2013*) |
| Calcium transient time to peak, duration, and amplitude | (*Coppini et al., 2013*) |
| I-V relationship and steady-state inactivation of L-type calcium current | (*Magyar et al., 2000*) |
| Sodium blockade is negatively inotropic | (*Gottlieb et al., 1990*; *Tucker et al., 1982*; *Legrand et al., 1983*; *Bhattacharyya and Vassalle, 1982*). |
| L-type calcium current blockade shortens the action potential | (*O'Hara et al., 2011*) |
| Early depolarisation formation under hERG block | (*Guo et al., 2011*) |
| Alternans formation at rapid pacing | (*Koller et al., 2005*) |
| Conduction velocity of ca. 65 m/s | (*Taggart et al., 2000*) |

**Validation**

| | |
|---|---|
| Action potential accommodation | (*Franz et al., 1988*) |
| S1-S2 restitution | (*O'Hara et al., 2011*) |
| Drug blocks and action potential duration | (*Dutta et al., 2017a*; *O'Hara et al., 2011*) |
| Hyperkalemia promotes postrepolarisation refractoriness | (*Coronel et al., 2012*) |
| Hypertrophic cardiomyopathy phenotype | (*Coppini et al., 2013*) |
| Drug safety prediction using populations of models | (*Passini et al., 2017*) |
| Physiological QRS and QT intervals in ECG | (*Engblom et al., 2005*; *van Oosterom et al., 2000*; *Bousseljot et al., 1995*; *Goldberger et al., 2000*) |

well as the $I_{Na}$, $I_{Kr}$, $I_{K1}$ and chloride currents. The multiobjective genetic algorithm optimisation was repeated several times, throughout the introduction of structural changes to the model. Once simulations with an optimised model fulfilled all calibration criteria, validation was conducted through evaluation against additional experimental recordings for drug block, disease, tissue and whole-ventricular simulations.

Details concerning the simulations are given in Appendix 1-15.1, namely the description of simulation protocols and ionic concentrations used (Appendix 1-15.1.1), representation of heart disease (Appendix 1-15.1.2), 1D fibre simulations (Appendix 1-15.1.3), population-of-models and drug safety assessment (Appendix 1-15.1.4), transmurality and whole-heart simulations with ECG extraction (Appendix 1-15.1.5), and a technical note on the update to the Matlab ODE solver which facilitates efficient simulation of the multiobjective GA (Appendix 1-15.1.6). Unless specified otherwise, the baseline ORd model (*O'Hara et al., 2011*) was used for comparison with the ToR-ORd model.

## ToR-ORd model structure

The ToR-ORd model follows the general ORd structure (*Figure 1A*). The cardiomyocyte is subdivided into several compartments: main cytosolic space, junctional subspace, and the sarcoplasmic reticulum (SR, further subdivided into junctional and network SR). Within these compartments are placed ionic currents and fluxes described by Hodgkin-Huxley equations or Markov models. The main ionic current formulations altered compared to ORd are highlighted in orange in *Figure 1A*.

## In-depth revision of the L-type calcium current

The $I_{CaL}$ current was deeply revisited, particularly with respect to its driving force, based on biophysical principles. This reformulation is of relevance to almost all models of cardiac electrophysiology.

The $I_{CaL}$ formulation in the ORd model is based on Hodgkin-Huxley equations, with the total current being a product of three components: 1) Open channel permeability, 2) A set of gating variables determining the fraction of channels being open, 3) The electrochemical driving force which acts on ions to move through the open channel based on the membrane potential and ionic concentrations on both sides of the membrane (more details in Appendix 1-5). In most Hodgkin-Huxley models of cardia currents, the driving force is computed as (V-$E_{ion}$), that is, the membrane potential minus equilibrium potential, either computed from the Nernst equation, or measured experimentally. However, starting with the Luo-Rudy model (LRd) of 1994 (*Luo and Rudy, 1994*), the driving force of ions via $I_{CaL}$ in cardiac models is modelled based on the Goldman-Hodgkin-Katz (GHK) flux equation. The driving force based on the GHK equation is:

$$\varphi_{CaL} = z^2 \cdot \frac{V \cdot F^2}{R \cdot T} \cdot \frac{[S]_i \cdot e^{\frac{z \cdot V \cdot F}{R \cdot T}} - [S]_o}{e^{\frac{z \cdot V \cdot F}{R \cdot T}} - 1},$$

where $z$ is the charge of the given ion, $V$ is the membrane potential, $F, R, T$ are conventional thermodynamic constants, and $[S]_i$, $[S]_o$ are intracellular and extracellular activities of the given ionic specie. $[S] = \gamma \cdot m$, where $\gamma$ is the ionic activity coefficient and $m$ the concentration (in either the intracellular or extracellular space, yielding $[S]_i$ or $[S]_o$).

### Determining ionic activity coefficients

In order to compute the ionic driving force via the GHK equation, it is necessary to know the ionic activity coefficients of the intracellular ($\gamma_i$) and extracellular ($\gamma_o$) space. The Luo-Rudy model and other Rudy-family models use $\gamma_o = 0.341$ for extracellular space and $\gamma_i = 1$ for the intracellular space. Models based on the Shannon model (*Shannon et al., 2004*) use 0.341 for both intracellular and extracellular space, but we were unable to find the motivation for this change.

The Debye-Hückel theory is commonly used to compute the activity coefficients. We used the Davies equation, which extends the basic Debye-Hückel equation to be accurate for ionic concentrations found in living cells (*Mortimer, 2008*):

$$\log \gamma_i = -A \cdot z_i^2 \cdot \left( \frac{\sqrt{I}}{1 + \sqrt{I}} - 0.3 \cdot I \right),$$

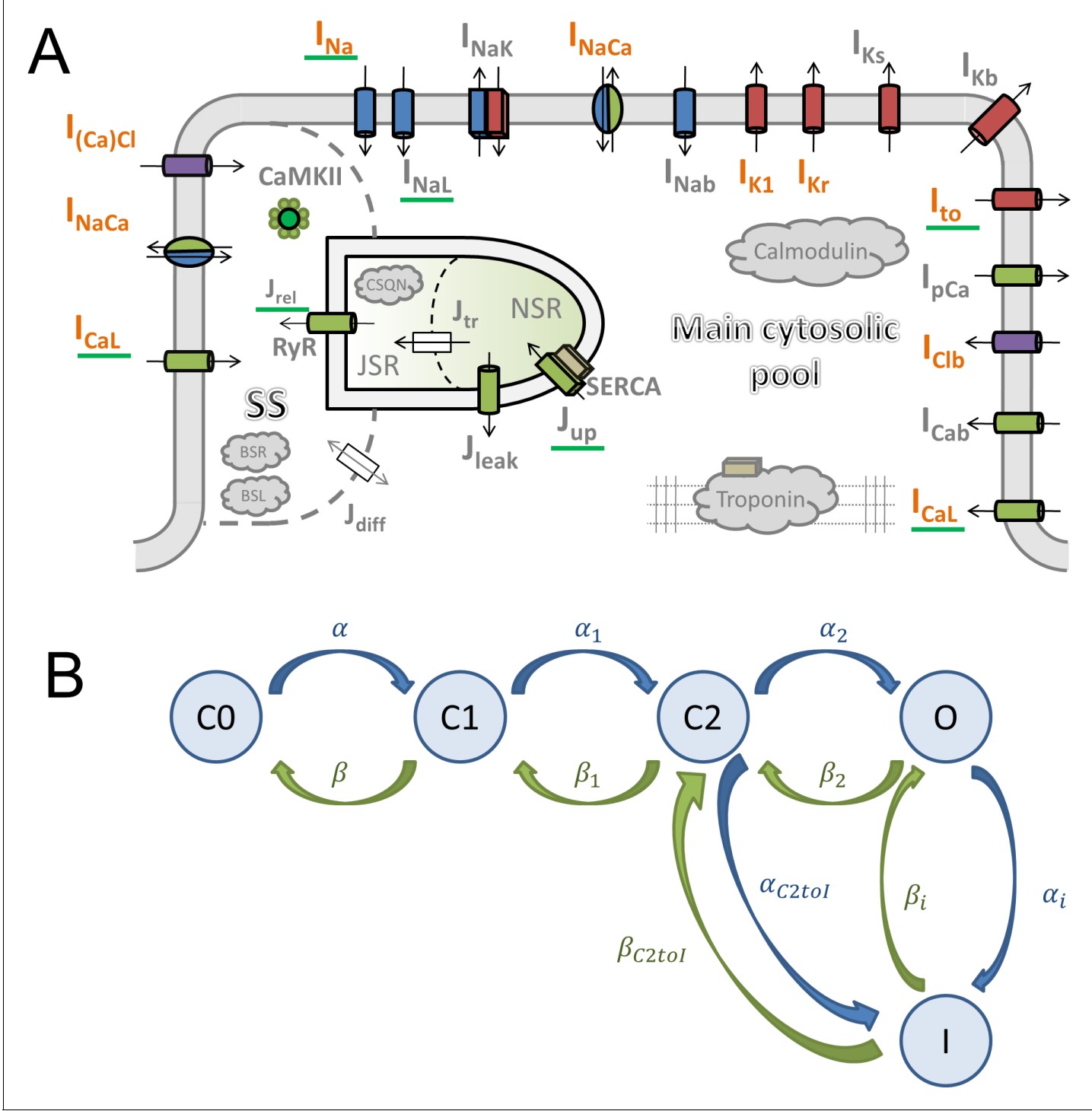

**Figure 1.** Model structure. (**A**) A schematic of the novel human ventricular myocyte model for electrophysiology and calcium handling. Orange indicates components, substituted, or added, compared to the original ORd model. 'SS' indicates junctional subspace compartment, where calcium influx via L-type calcium current occurs and where calcium is released from the sarcoplasmic reticulum. 'JSR' and 'NSR' are junctional and network sarcoplasmic reticulum compartments, respectively. 'Main cytosolic pool' is the remaining intracellular space. Transmembrane currents are indicated with an 'I' in their name, with fluxes indicated as 'J'. Components with a green underscore are modulated by CaMKII signalling. (**B**) The structure of the Lu-Vandenberg (*Lu et al., 2001*) Markov model used for the rapidly activating delayed rectifier repolarisation current (I_Kr). The transition rates are given in Appendix 1-15.3.5.

where $A$ is a constant (~0.5 for water at 25°C, ~0.5238 at 37°C), $z_i$ is the charge of the respective ion, and $I$ is the ionic strength of the solution. The ionic strength is defined as:

$$I = 0.5 \cdot \sum_i m_i \cdot z_i^2,$$

where $m_i$ is the concentration of the $i$-th ionic specie present. For concentrations in a study measuring properties of $I_{CaL}$ (*Magyar et al., 2000*), $I$ is ca. 0.15-0.17. This warrants the use of Davies equation, which was shown to be accurate for $I$ up to 0.5, unlike the basic Debye-Hückel equation, which is accurate for $I$ up to 0.01 only (*Mortimer, 2008*).

We implemented the computation of ionic coefficients based on the Davies equation dynamically, so that the activity coefficients are estimated at every simulation step. This allows accurate representation of the driving force when ionic concentrations are disturbed, such as at varying pacing rates, or during homeostatic imbalance. The dynamic computation is also used to estimate ionic activity coefficients for potassium and sodium flowing through the calcium channels, taking into account their different charge.

Throughout our simulations, both intracellular and extracellular activity coefficients generally lie between 0.61 and 0.66. Importantly, this estimate shows that the intracellular and extracellular activity coefficients are relatively similar (corresponding to the broadly similar total concentration of charged molecules), in contrast with the original values. Particularly, the origin of the intracellular activity coefficient $\gamma_i = 1$ in the Luo-Rudy model is unclear, as by the Davies (or by any Debye-Hückel variant) equation, $I$ would have to be zero, which is possible only when there are no ions present.

## Activation curve extraction

An additional improvement in the $I_{CaL}$ formulation is the estimation of its activation curve. In brief, we implement a consistent use of the GHK equation for the extraction of the activation curve and for the $I_{CaL}$ formulation in the ToR-ORd model. The activation curve is obtained via dividing the experimentally measured I-V relationship of the current by the expected driving force for each pulse potential (see Appendix 1-3 for a graphical overview of the process). However, we identified a theoretical inconsistency in previous cardiac models across species (e.g. *Luo and Rudy, 1994*; *Hund et al., 2008*; *O'Hara et al., 2011*; *Shannon et al., 2004*; *Grandi et al., 2010*; *Carro et al., 2011*): whereas the Nernstian driving force of (V-$E_{Ca}$) is used to derive the activation curve, the GHK driving force is then used to calculate $I_{CaL}$. Indeed, experimental studies reporting the activation curve of $I_{CaL}$ generally use the Nernstian driving force of (V-$E_{Ca}$) with $E_{Ca}$ being the experimentally measured reversal potential of approximately 60 mV. This is explicitly stated in *Linz and Meyer (2000)*, and also the activation curve by *Magyar et al. (2000)* used in the ORd model is consistent with dividing the IV relationship with (V-60).

In this study, we propose that, for consistency, the same equation needs to be applied both to obtain the activation curve from the I-V curve and to represent the driving force in the current formulation. Thus, in the ToR-ORd model, the activation curve for $I_{CaL}$ was obtained by dividing the I-V curve from *Magyar et al. (2000)* by the GHK-based driving force, computed using ionic activity coefficients based on the Davies equation (as explained in the previous Section) and intracellular and extracellular ionic concentrations as in *Magyar et al. (2000)*. The following capped Gompertz function (a flexible sigmoid) was found to be the best fit to the resulting steady-state activation curve:

$$d_\infty = \left( 1.0763 \cdot e^{-1.007 \cdot e^{-0.0829 \cdot V}} \text{ for } V \leq 31.4978 \,|\, 1 \text{ otherwise} \right),$$

where $V$ is the membrane potential.

## Other $I_{CaL}$ changes

20% of $I_{CaL}$ was placed in the main cytosolic space, consistent with the literature (*Scriven et al., 2010*). This increases the plateau-supporting capability of $I_{CaL}$, given that the myoplasmic $I_{CaL}$ is subject to a weaker calcium-dependent inactivation than $I_{CaL}$ in the junctional subspace. Other minor changes are given in Appendix 1-15.3.3.

## $I_{Kr}$ replacement

The calibration of the ToR-ORd model's AP morphology to experimental data resulted in problematic response to calcium blockade during an early phase of the model development when the original $I_{Kr}$ formulation was used (further details in Appendix 1-12). $I_{CaL}$ block is known to shorten APD experimentally (*O'Hara et al., 2011*) but resulted in a major APD prolongation in simulations instead. This discrepancy could not be resolved through parameter optimisation. A mechanistic analysis revealed that this follows from the lack of ORd $I_{Kr}$ activation, which is however not consistent with relevant experimental data (*Lu et al., 2001*). We therefore considered alternative $I_{Kr}$ formulations and specifically the Lu-Vandenberg (*Lu et al., 2001*) Markov model (*Figure 1B*). The Lu-Vandenberg $I_{Kr}$ model is based on extensive experimental data allowing the dissection of activation and recovery from inactivation and provided the best agreement with experimental data, specifically when considering the AP plateau potentials reported experimentally. In Appendix 1-12, we: (1) provide a detailed explanation of origins of AP prolongation following $I_{CaL}$ block in a model which manifests experimental data-like plateau potentials and which contains the ORd $I_{Kr}$ formulation; (2) explain why this phenomenon occurs only in a model with experimental data-like plateau potentials, but not in the original high-plateau ORd model; (3) compare the ORd and Lu-Vandenberg $I_{Kr}$ formulations with experimental data, demonstrating the good agreement with experimental data of the Lu-Vandenberg formulation but not the ORd.

Following the inclusion of the Lu-Vandenberg $I_{Kr}$ formulation, all models generated during model calibration exhibited APD shortening in response to $I_{CaL}$ block.

## Changes in $I_{Na}$, $I_{(Ca)Cl}$, $I_{Clb}$ and $I_{K1}$

The $I_{Na}$ current formulation was replaced by an alternative human-based formulation (*Grandi et al., 2010*), given established limitations of the original model with regards to conduction velocity and excitability (*O'Hara et al., 2011*, comment on article from 05 Oct 2012). The Grandi $I_{Na}$ model was updated to account for CaMKII phosphorylation (Appendix 1-15.3.1).

Also from the Grandi model, we added the calcium-sensitive chloride current $I_{(Ca)Cl}$ and background chloride current $I_{Clb}$ formulation (*Grandi et al., 2010*). Neither model was changed compared to the original formulations, but the intracellular concentration of $Cl^-$ was slightly increased (Appendix 1-15.1.1). In accordance with recent observations, $I_{(Ca)Cl}$ was placed in the junctional subspace (*Magyar et al., 2017*). The motivation to add these currents was to facilitate the shaping of post-peak AP morphology (via $I_{(Ca)Cl}$), with $I_{Clb}$ playing a dual role stemming from its reversal potential of ca. $-50$ mV. It slightly reduces plateau potentials during the action potential, but during the diastole, it depolarises the cell slightly, improving the reaction to $I_{K1}$ block as explained in the next subsection.

The $I_{K1}$ model was replaced with the human-based formulation by *Carro et al. (2011)*, as it was shown to be key for simulations of hyperkalemic conditions. The $I_{K1}$ replacement was done before hyperkalemia simulation, not violating the classification of hyperkalemia criterion as a validation step. Extracellular potassium concentration in a healthy cell was reduced from 5.4 to 5 mM to fall within the physiological range (*Zacchia et al., 2016*).

## Calibration of $I_{K1}$ block and resting membrane potential

When evaluating the baseline ORd model against the selected criteria, we observed that a reduction in $I_{K1}$ results in hyperpolarisation of the cell (from $-88$ to $-88.16$ mV at 1 Hz pacing). However, it is established that $I_{K1}$ reduction depolarises cells experimentally (*Dhamoon and Jalife, 2005*). Changes made during ToR-ORd calibration (predominantly the altered balance of currents during diastole and the inclusion of background chloride current) result in ToR-ORd manifesting depolarization in response to $I_{K1}$ block, consistent with experimental data.

## Multiobjective genetic algorithm

We applied a multiobjective genetic algorithm (MGA, @gamultiobj function in Matlab, *Deb, 2001*) to automatically re-fit various model parameters. Based on preliminary experimentation, we used a two-dimensional fitness. We used MGA rather than an ordinary genetic algorithm or particle swarm optimisation, given that MGA optimises towards a Pareto front rather than a single optimum, implicitly maintaining population diversity. The Pareto front is the set of all creatures which are not

*dominated* by any other creature in the population, that is creatures for which there is no other creature better in all fitness dimensions. Therefore, a subpopulation of diverse solutions is maintained, and the optimiser consequently has less of a tendency to converge to a single local optimum compared to single-number fitness approaches. In addition, the crossover operator of GA is well suited for a task where multiple criteria are optimised, given that creatures in the population may efficiently share partial solutions to various subcriteria. The fitness used in this study is described in greater detail in Appendix 1-1.

## Evaluation pipeline and code

To facilitate the model validation and future work, we also provide an automated 'single-click' evaluation pipeline. It runs automatic simulations to extract and visualise single-cell biomarkers including those related to AP morphology, effect of key channel blockers, early afterdepolarisations (EAD), and alternans measurement. The pipeline generates a single HTML report containing all the results; see Appendix 1-15.2 for a visualisation. The code for our model (Matlab and CellML), the validation pipeline, and the experimental data on human AP morphology are available at https://github.com/jtmff/torord (*Tomek, 2019*; copy archived at https://github.com/elifesciences-publications/torord). An informal blog giving further insight into the choices we made, as well as general thoughts on the development of ToR-ORd and computer models in general, is available at https://underlid.blogspot.com/.

We designed the Matlab code used to simulate our model so that the simulation core is structured into functions computing currents, making the high-level organisation of code clear, and facilitating inclusion of alternative current formulations. In addition, a CellML file encoding our model is also provided. This makes the model readily runnable in several simulators in addition to Matlab (e.g. Chaste [*Pitt-Francis et al., 2009*] and OpenCOR [*Garny and Hunter, 2015*]). Furthermore, the Myokit library (*Clerx et al., 2016*) enables conversion of the CellML file to other languages (such as C or Python).

## Results

## Calibration based on AP, calcium transient, and L-type calcium current properties

The AP morphology of the ToR-ORd is within or at the border of the interquartile range of the Szeged-ORd experimental data (*Figure 2A*). This is a major improvement compared to the original ORd morphology, which overestimates plateau potentials, particularly during early plateau (*Figure 2A*). The fact that the early plateau potential is around 20–23 mV is clearly apparent from experimental recordings and is further corroborated by additional studies in human tissue samples (*Jost et al., 2013*, Figure 6) and isolated human cardiomyocytes (*Coppini et al., 2013*). We note that compared to the Szeged-ORd dataset (*Britton et al., 2017*), our model manifests a slightly increased peak membrane potential in the single-cell form, similar to single-cell experimental data (*Coppini et al., 2013*). This is a design choice related to the fact that the Szeged-ORd dataset contains recordings of small tissue samples, which are expected to manifest a reduced peak potential compared to single-cell. When coupled in a fibre, ToR-ORd manifests conduction velocity of 65 cm/s, which is consistent with clinical data (*Taggart et al., 2000*).

Both time to peak calcium and duration of calcium transient at 90% recovery obtained with the ToR-ORd model are within the standard deviation of experimental data in isolated human myocytes (*Coppini et al., 2013*), whereas ORd slightly overestimated the calcium transient duration (*Figure 2B*). The calcium transient amplitude of ToR-ORd also matches the Coppini et al. data after accounting for the different APD (Appendix 1-8).

As described in Materials and methods, the ToR-ORd $I_{CaL}$ activation curve was extracted from experimental data, using the Goldman-Hodgkin-Katz formulation of ionic driving force, ensuring theoretical consistency, unlike the ORd $I_{CaL}$ formulation (*Figure 2C*). This considerably improves the results of simulated protocols to obtain IV relationship (*Figure 2D*), validating the theory-driven changes (see Appendix 1-4 for the demonstration of how the updated activation curve underlies the improvement). The simulation of the protocol measuring steady-state inactivation also reveals improved agreement of ToR-ORd with experimental data compared to ORd (*Figure 2E*). The

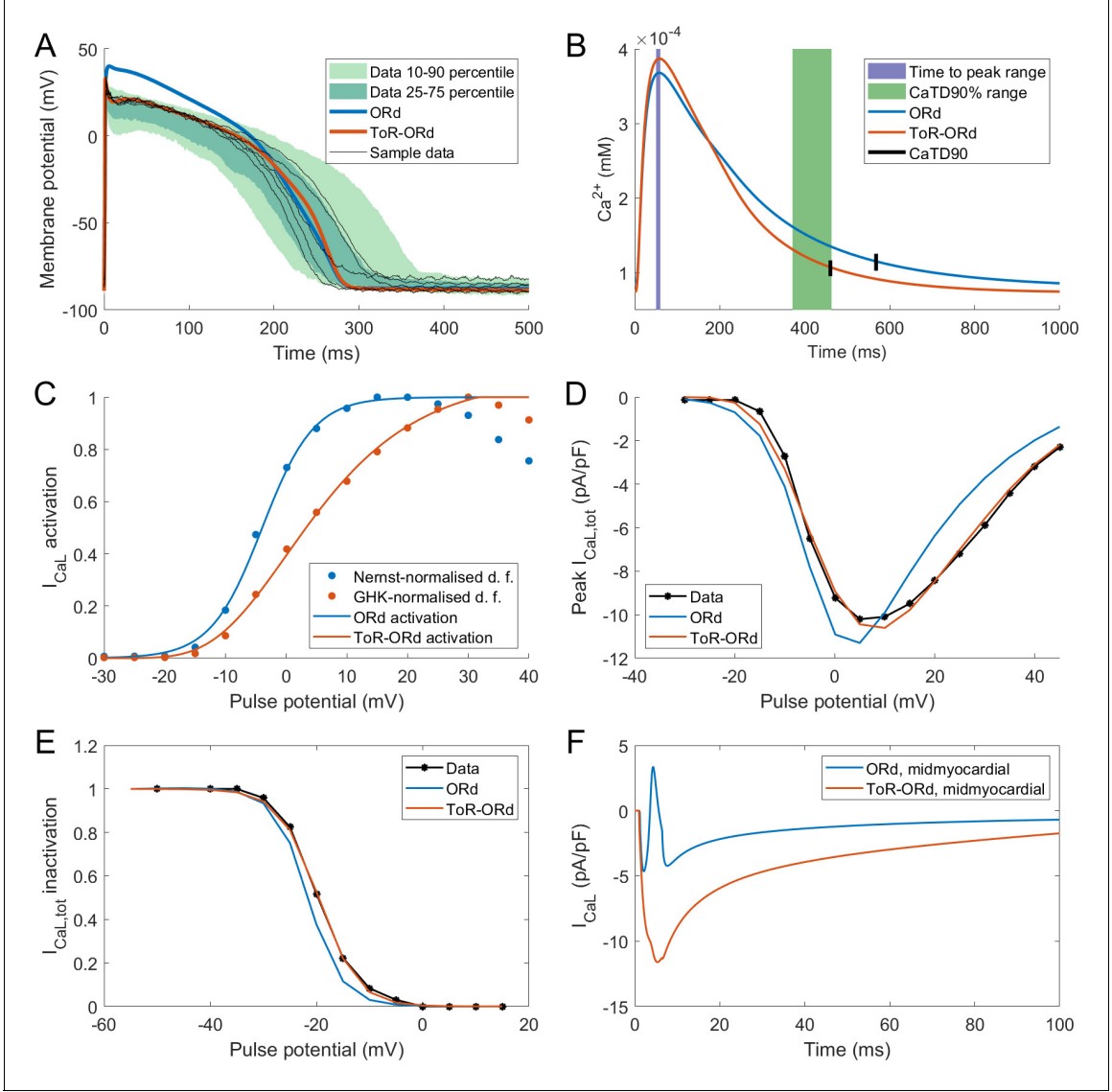

**Figure 2.** Action potential, calcium transient, and $I_{CaL}$ in ToR-ORd. Action potential (A) and calcium transient (B) at 1 Hz obtained with the ToR-ORd model following calibration, compared to those obtained with the ORd model and experimental data from *O'Hara et al. (2011)* and *Coppini et al. (2013)*, respectively. The purple and green zones in (B) stand for mean ± standard deviation. The duration of calcium transient at 90% recovery was extracted from figures in *Coppini et al. (2013)*, adding the time to peak and time from peak to 90% recovery. (C) Activation curves used in the ToR-ORd and ORd models (blue and red lines, respectively). The points correspond to the IV relationship measured in *Magyar et al. (2000)* normalised by the Nernstian driving force (d.f.) assuming reversal potential of +60 mV (blue points) and the GHK driving force (red points). (D, E) I-V relationship and steady-state inactivation as measured in ToR-ORd (red line) versus ORd model (blue line) versus experimental data from *Magyar et al. (2000)* (black points with line). $I_{CaL,tot}$ is the sum of currents corresponding to all ions ($Ca^{2+}$, $Na^+$, and $K^+$) passing through the L-type calcium current channels. (F) L-type calcium current of a midmyocardial cell, showing current reversal in ORd, but not in ToR-ORd. Only the calcium component of $I_{CaL,tot}$ is shown to demonstrate that the current reversal is not due to other ions. We note that the difference in total amplitude of $I_{CaL}$ in *Figure 2F* follows predominantly from different action potential shape in ToR-ORd vs ORd, consistent with the I-V relationship.

difference between measured ORd steady state inactivation and the experimental data (ca. two times stronger inactivation at around −15 mV, which is relevant for EAD formation) is initially surprising, given that the equation of ORd $I_{CaL}$ steady-state inactivation curve provides a good fit to the same experimental data. This difference follows from the formulation of calcium-dependent inactivation of $I_{CaL}$ (see Appendix 1-5 for details).

We observed that in cases of elevated $I_{CaL}$ (e.g. in midmyocardial cells), ORd reverses current direction towards positive values, which is an unexpected behaviour given its reversal potential of 60

mV. Conversely, the ToR-ORd model yields negative $I_{CaL}$ values in such conditions, consistent with it being an inward current (*Figure 2F*). This is a direct consequence of the updates to the extracellular/intracellular calcium activity coefficients (as explained in Appendix 1-6), which supports their credibility and it is important for cases of elevated $I_{CaL}$, such as under ß-adrenergic stimulation.

We have also simulated a P2/P1 protocol as measured experimentally by *Fülöp et al. (2004)*, where two rectangular pulses are applied with varying interval between them. Both ORd and ToR-ORd qualitatively agree with the experimental data (Appendix 1-7).

## Calibration: inotropic effects of sodium blockers

*Figure 3A-D* illustrates AP and calcium transient changes caused by block of sodium currents in ToR-ORd (left) and ORd (right). As sodium blockers act on channel $Na_v1.5$ mediating both the fast ($I_{Na}$) and late ($I_{NaL}$) sodium current (*Makielski, 2016*), we simulate the effect of combined partial $I_{Na}$ and $I_{NaL}$ block. The ToR-ORd model manifests a small reduction in calcium transient amplitude (*Figure 3C*), unlike ORd, which gives a sizeable increase (*Figure 3D*); ToR-ORd is thus consistent with the observed negative inotropy of sodium blockers (*Gottlieb et al., 1990*; *Tucker et al., 1982*; *Legrand et al., 1983*; *Bhattacharyya and Vassalle, 1982*). This is a major improvement in the ToR-ORd model, as sodium current reduction is involved in a range of disease conditions in addition to pharmacological block.

Experimental evidence shows that the ratio of $I_{Na}$ and $I_{NaL}$ block is drug and dose-dependent, with $I_{NaL}$ usually being blocked more than $I_{Na}$ (Appendix 1-9). *Figure 3E,F* illustrates the change in calcium transient amplitude obtained with the ToR-ORd and ORd models, respectively, for several combinations of $I_{Na}$ and $I_{NaL}$ availability. Both models show a similar general trend where reduced $I_{Na}$ availability increases calcium transient amplitude and reduced $I_{NaL}$ availability diminishes it; however, the models differ strongly in relative contributions of these components. The ToR-ORd model shows negative inotropy for almost all combinations of blocks. A mild increase in inotropy may be achieved only under near-exclusive $I_{Na}$ block. Conversely, ORd shows a general tendency for increased calcium transient amplitude; a reduction occurs only when the sodium current block targets near-exclusively $I_{NaL}$. ToR-ORd presents a greater calcium transient amplitude reduction than ORd in response to $I_{NaL}$ block, as the current has a greater role in indirect modulation the cell's calcium loading via APD change. At the same time, ToR-ORd shows a much smaller calcium transient amplitude increase in response to $I_{Na}$ block than ORd because of the updated $I_{CaL}$ activation curve (*Figure 2C*), as well as closer-to experimental data AP morphology (*Figure 2A*) and its effect on $I_{CaL}$. A detailed explanation is given in Appendix 1-10.

Fibre simulations carried out to assess the effect of cell coupling on the effect of sodium block are consistent with the single-cell simulations (Appendix 1-11). The difference in response to half-block of $I_{Na}$ and $I_{NaL}$ between ToR-ORd and ORd is even larger, as ToR-ORd in fibre predicts a greater reduction in calcium transient amplitude than in single cell (−14% vs −6% respectively), while ORd in fibre predicts a slightly greater increase in calcium transient amplitude than in single cell (+25% vs +24%).

With the ToR-ORd model, the 14% reduction in CaT amplitude in the electrically coupled fibre with 50% block of both $I_{Na}$ and $I_{NaL}$ is generally consistent with clinical data on sodium blockers: Encainide reduced stroke work index by 15% and cardiac index by 8% (*Tucker et al., 1982*). In another study using encainide, the cardiac index was reduced by 18% and the stroke volume index by 28% (*Gottlieb et al., 1990*). Flecainide reduced left ventricular stroke index by 12% and the left ventricular ejection fraction by 9% (*Legrand et al., 1983*). Simulations with the ToR-ORd model show overall agreement with the clinical data. However, a direct quantitative comparison is challenging given the different indices of contractility measured (CaT amplitude versus clinical indices) and that it is not possible to estimate the exact ratios of $I_{Na}$ and $I_{NaL}$ block in clinical data (Appendix 1-9).

## Calibration: proarrhythmic behaviours (alternans and early afterdepolarisations)

EADs are an important precursor to arrhythmia, manifesting as a membrane potential depolarisation during late plateau and/or early repolarisation. They are thought to arise mainly from $I_{CaL}$ current reactivation (*Weiss et al., 2010*). The ToR-ORd model manifests EADs at conditions used experimentally in nondiseased human endocardium (*Guo et al., 2011*; *Figure 4A*). The amplitude of

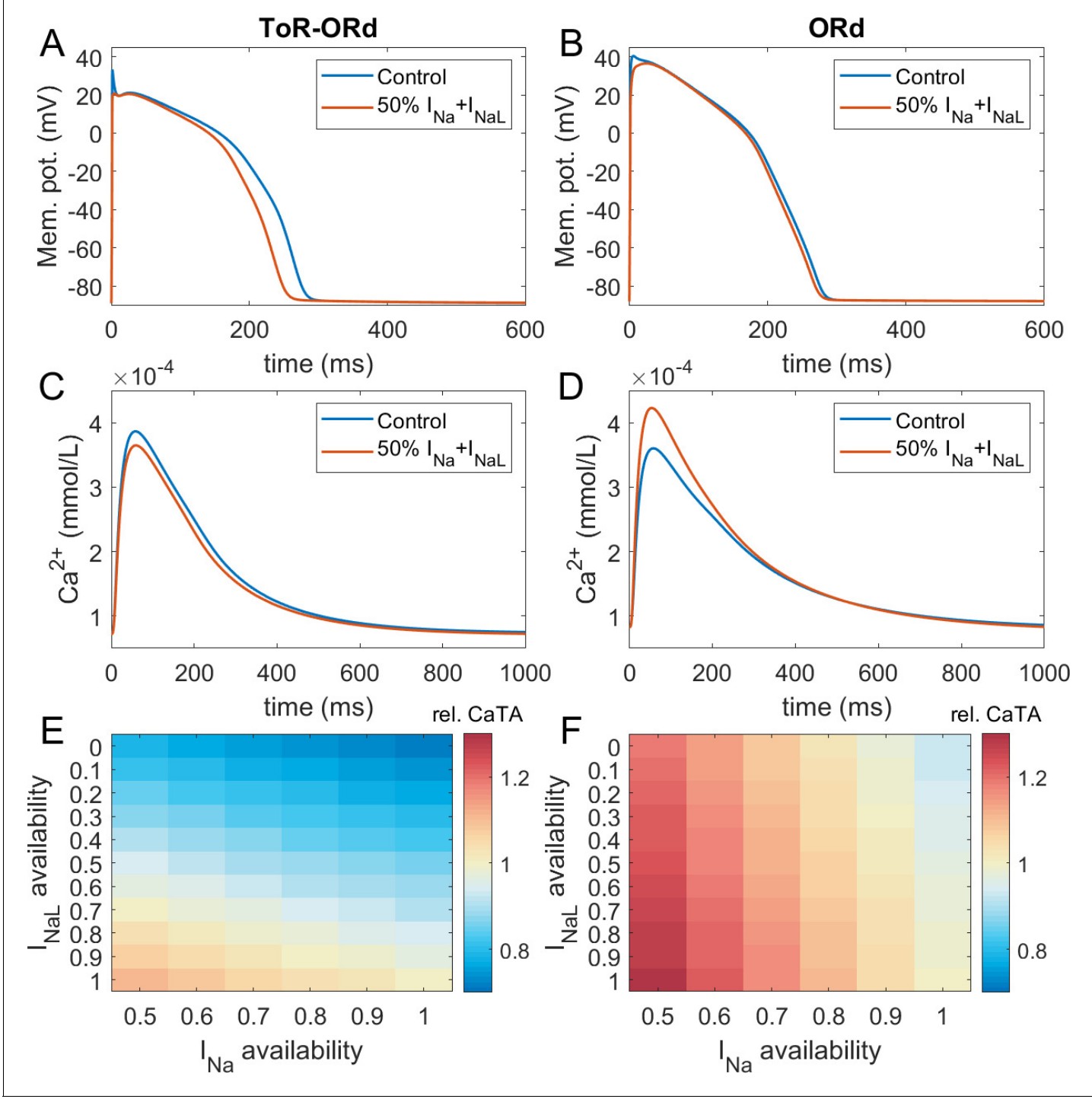

**Figure 3.** Sodium blockade in ToR-ORd and ORd. Simulated effect of sodium current block on the action potential (**A, B**) and calcium transient (**C,D**) using the ToR-ORd (left) and ORd (right) models. Control simulations are shown as blue traces, whereas results for 50% block of $I_{Na}$ and $I_{NaL}$ are shown as red traces. Panels (**E,F**) show values of changes in calcium transient amplitude with respect to control ('rel. CaTA') for varying degrees of $I_{Na}/I_{NaL}$ block for ToR-ORd and ORd respectively (one is no change, values > 1 correspond to an increase the calcium transient amplitude).

simulated EADs is 14 mV (*Figure 4B*), which matches the maximum EAD amplitude shown by *Guo et al. (2011)*. We also note that the experimental data by Guo et al. manifest early plateau potential of ca. 23 mV (which is matched by ToR-ORd), in line with other studies we referred to previously regarding this matter.

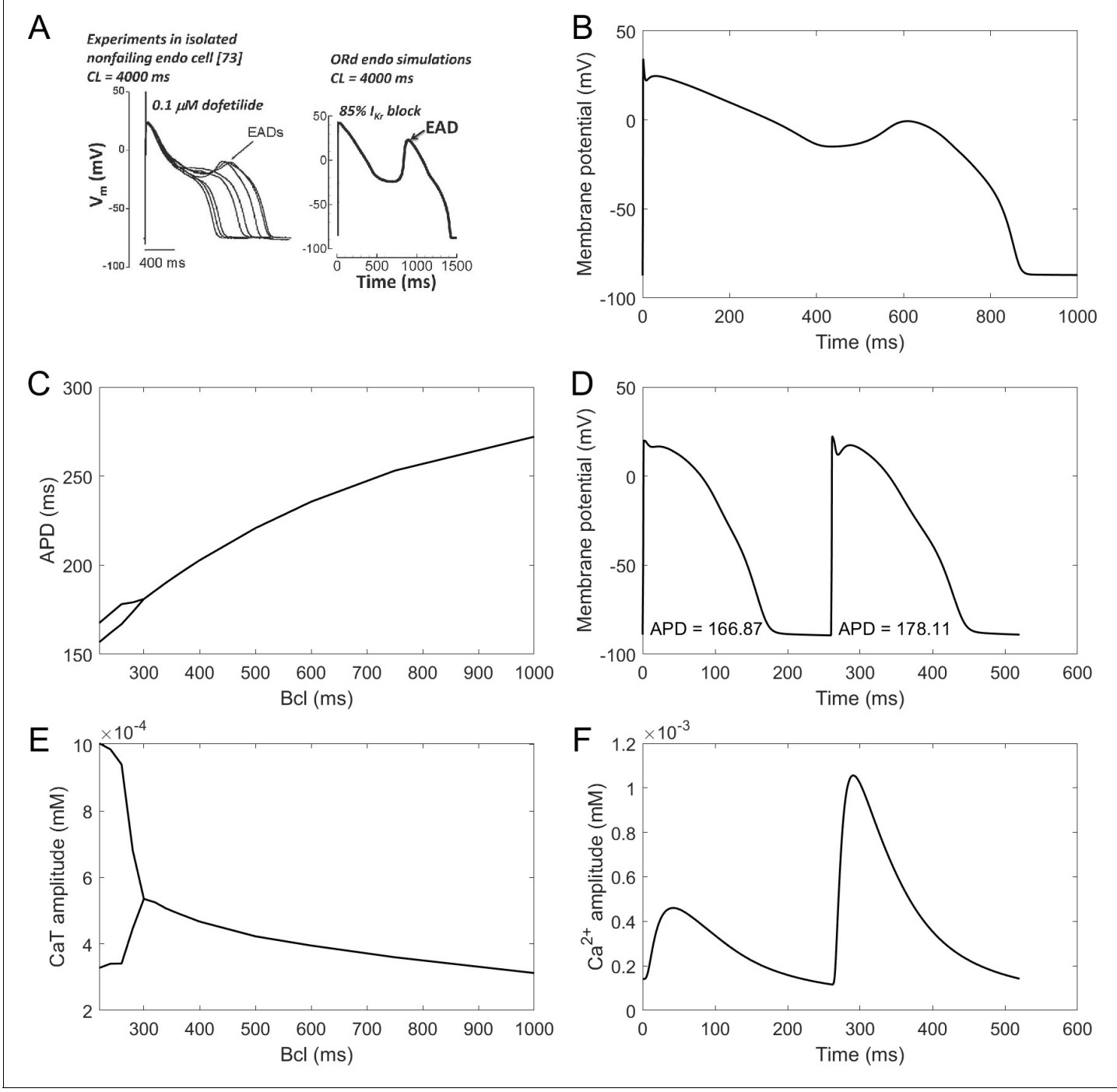

**Figure 4.** EADs and alternans. (**A**) Experimentally observed EADs produced in nondiseased human myocytes exposed to 0.1 µM dofetilide (~85% I_Kr block) paced at 0.25 Hz (*Guo et al., 2011*) and the corresponding simulation using the ORd model (*O'Hara et al., 2011*), figure reproduced as allowed by the CC-BY license). See Appendix 1-15.1.1 for ionic concentrations used. (**B**) An EAD produced by the ToR-ORd model at the same conditions. (**C,E**) APD and calcium transient amplitude versus pacing base cycle length (bcl); the bifurcation indicates alternans. (**D,F**) Example of APD and calcium transient alternans at the bcl of 260 ms.

Repolarisation alternans is another established precursor to arrhythmia, facilitating the formation of conduction block (*Weiss et al., 2006*). It is induced by rapid pacing and it is mostly thought to arise from calcium transient amplitude oscillations being translated to APD oscillations (*Pruvot et al., 2004*), although purely voltage-driven mechanism was also proposed (*Nolasco and*

*Dahlen, 1968*). Alternans in the ToR-ORd model is calcium-driven and appears via the same mechanism as in the ORd: sarcoplasmic reticulum calcium cycling refractoriness (*Tomek et al., 2018*). It occurs at rapid pacing, in both calcium and APD (*Figure 4C–F*). The peak APD alternans amplitude (difference in APD between consecutive beats) is 12 ms, which is matches the value 11 ± 2 ms reported in human hearts without a structural disease (*Koller et al., 2005*). Direct quantitative comparison is however slightly limited by the fact that the data were recorded in RV septum, which may or may not differ from endocardial cells in alternans amplitude.

## Validation: drug-induced effects on rate dependence of APD

*Figure 5* illustrates simulations of drug action using the ToR-ORd model (red traces), compared to experimental data (black traces) and to simulations with the ORd model reparametrised

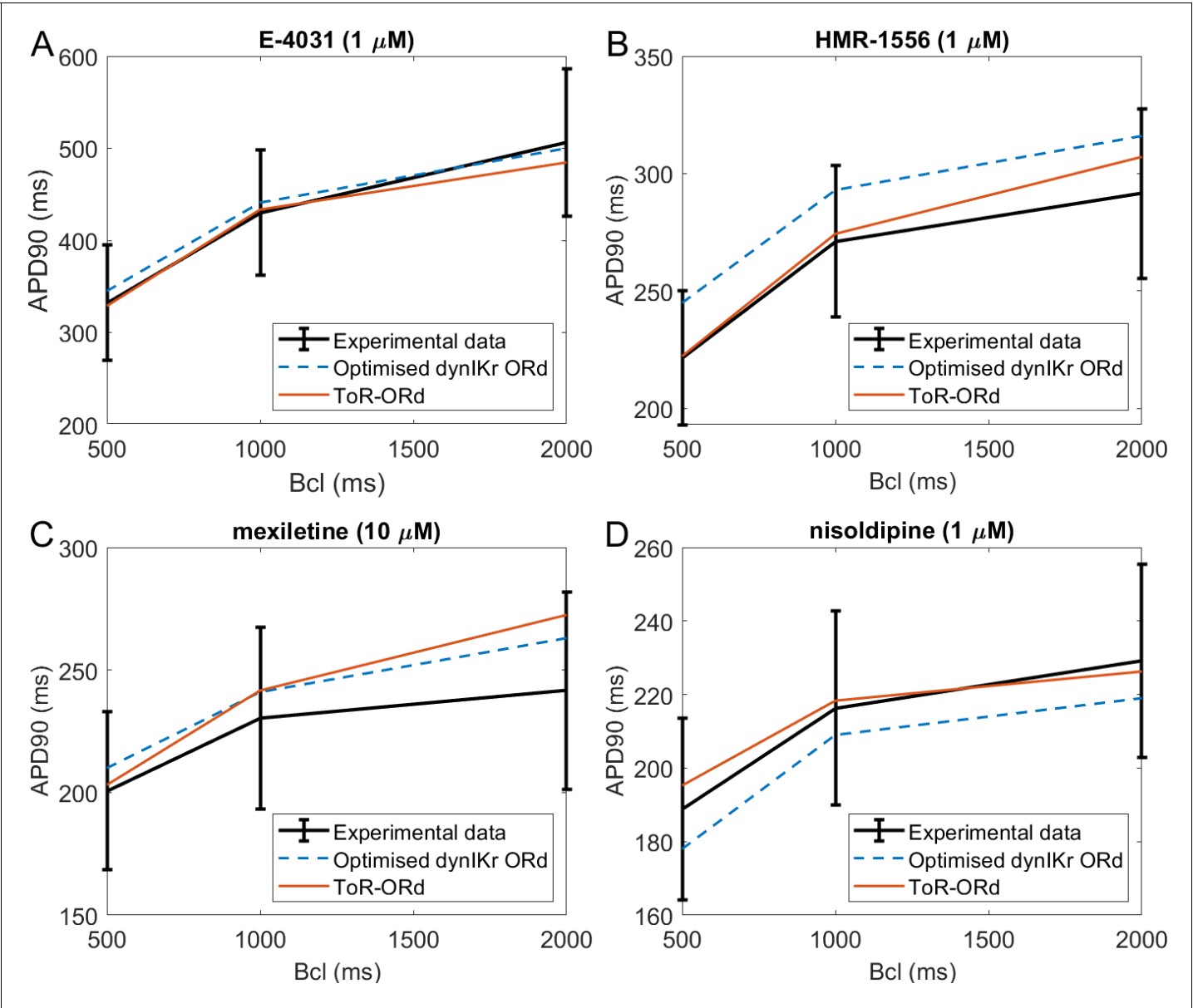

**Figure 5.** Drug block and APD. All four panels contain mean and standard deviation of experimental data (black) as measured in *O'Hara et al. (2011)* for three basic cycle lengths (bcl), predictions of the Dutta optimised dynamic-$I_{Kr}$ ORd model (blue, *Dutta et al., 2017a*), and the predictions of the ToR-ORd model (red). (A) 1 µM E-4031 (70% $I_{Kr}$ block). (B) 1 µM HMR-1556 (90% $I_{Ks}$ block). (C) 10 µM mexiletine (54% $I_{NaL}$, 9% $I_{Kr}$, 20% $I_{CaL}$ block). (D) 1 µM nisoldipine (90% $I_{CaL}$ block). Drug concentrations and their effects on channel blocks are based on *Dutta et al. (2017b)*.

by *Dutta et al. (2017a)* (blue dashed lines). APD is shown in the presence of $I_{Kr}$ block (E-4031, *Figure 5A*), $I_{Ks}$ block (HMR-1556 *Figure 5B*), multichannel block of $I_{NaL}$, $I_{CaL}$, $I_{Kr}$ (mexiletine, *Figure 5C*), and a $I_{CaL}$ block (nisoldipine, *Figure 5D*), at base cycle lengths of 500, 1000, and 2000 ms. We note that while the Dutta et al. model was specifically optimised for response of APD to these drug blocks, no such treatment was applied to the ToR-ORd model, making the results presented here an independent validation. Appendix 1-13 contains further details on the choice and use of the drug data.

The predictions produced by the ToR-ORd model are in good agreement with experimental data, particularly given the lack of optimisation towards this result. Simulating E-4031, ToR-ORd provides a prediction similar to the experimental data mean and the Dutta model (*Figure 5A*). This is crucial, given the key role of $I_{Kr}$ in the repolarisation reserve of human cardiomyocytes. The response to $I_{Ks}$ blockade via HMR-1556 is even better in ToR-ORd than in the Dutta model, which is also within standard deviation of the data, but carries a clear trend towards AP prolongation (*Figure 5B*). When simulating the multichannel blocker mexiletine, ToR-ORd prediction is within standard deviation of the experimental data, with the Dutta model giving similar or closer-to-mean predictions at 0.5 and 1 Hz (*Figure 5C*). The predicted effect of the calcium blocker nisoldipine in the ToR-ORd model matches well the experimental data mean (*Figure 5D*), even better than the Dutta model (also within standard deviation). We note that the good performance of the simulated nisoldipine effect critically relies on the $I_{Kr}$ replacement (Materials and methods and Appendix 1-12).

### Validation: APD accommodation and S1-S2 restitution

Experimental measurements in human cardiomyocytes (*Franz et al., 1988*; *Bueno-Orovio et al., 2012*) show how the APD shortens upon increase in pacing frequency, and then prolongs again, as

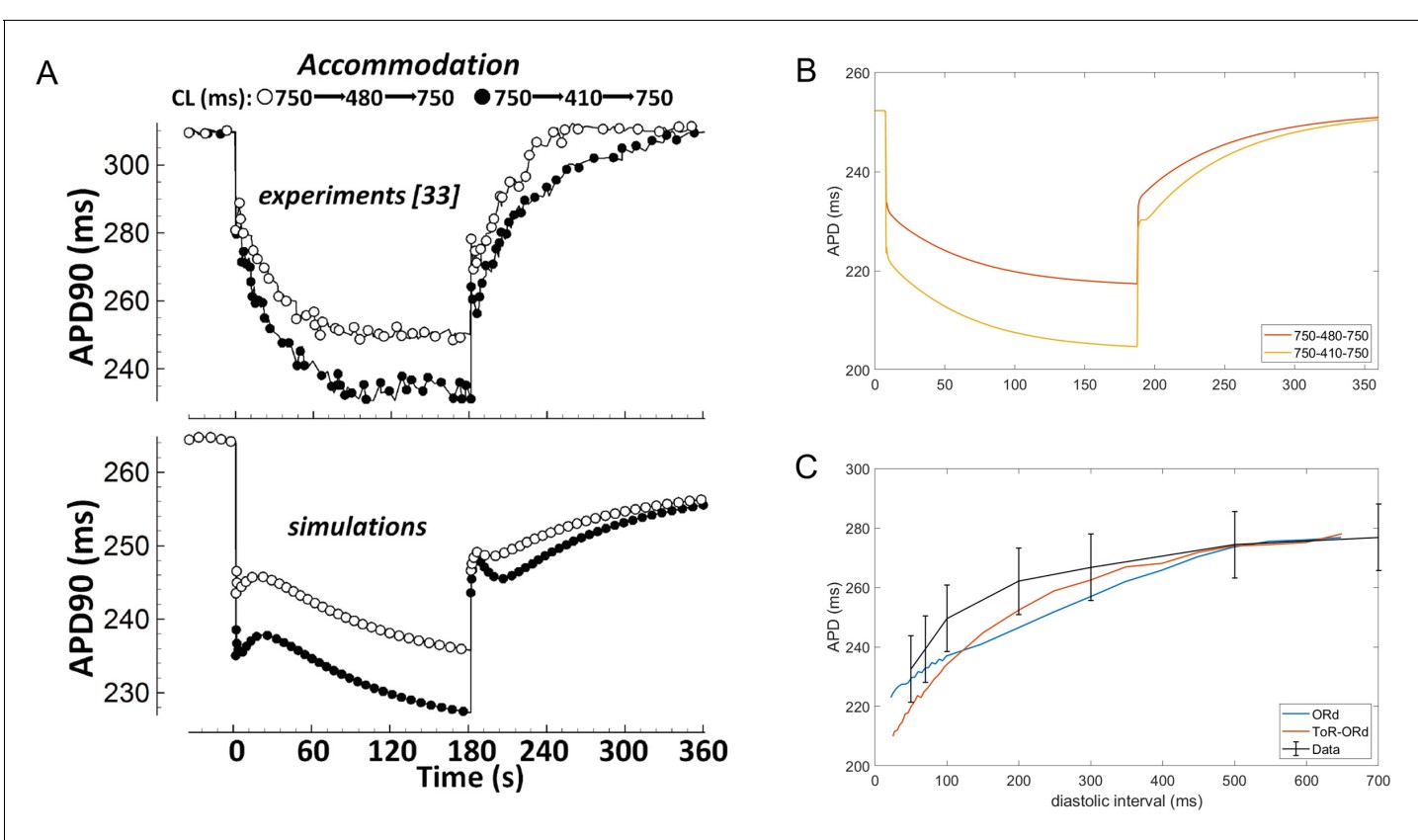

**Figure 6.** APD accommodation and S1-S2 protocol. (**A**) APD accommodation measured experimentally (*Franz et al., 1988*) and in simulation of the ORd model (reproduced as allowed by the CC-BY licence from *O'Hara et al., 2011*). (**B**) APD accommodation in ToR-ORd. (**C**) S1-S2 restitution curve (S1 = 1000 ms) in ToR-ORd fibre, ORd fibre (including $I_{Na}$ modification to facilitate propagation *Passini et al., 2016*) and human tissue samples data (*O'Hara et al., 2011*).

the pacing frequency returns to control (*Figure 6A*, top). APD adaptation dynamics with changes in heart rate are regulated by changes in sodium homeostasis (*Pueyo et al., 2011*), and their manifestation in QT adaptation have been shown to be useful for arrhythmia risk prediction (*Pueyo et al., 2004*). While simulations with the ORd model capture the general trend of APD accommodation, there are differences compared to the experimental data (*Figure 6A*). First, changes in pacing rate are followed by slow-dynamics (~30 s) APD prolongation not present in the experimental recordings. Second, the time constant of accommodation is generally slow. Conversely, the ToR-ORd model reproduces the pattern of accommodation well, where the change in APD soon after change in frequency is relatively fast, and then gradually slows down (*Figure 6B*). This suggests that the ionic balance in ToR-ORd is likely to have been improved compared to ORd.

A second indicator of how a model responds to a change in pacing frequency is the S1-S2 restitution protocol. The S1-S2 restitution curve obtained with the ToR-ORd model is given in Figure (*Figure 6C*), showing a good agreement with the experimental data (*O'Hara et al., 2011*).

## Validation: populations of models and drug safety prediction

Drug safety testing is one of the key applications of computer modelling which has yielded highly promising results (*Passini et al., 2017*). To assess the suitability of ToR-ORd for drug safety testing, we replicated the study by *Passini et al. (2017)*, which was carried out using populations of models based on the ORd model. Two populations were created based on ToR-ORd similarly to the original study, altering conductances of important currents within the ranges of 50–150% and 0–200%. Models in both populations are stable under significant perturbation of ionic conductances, which supports the robustness of the model (*Figure 7A*).

Prediction of the risk of drug-induced Torsades de Pointes based on simulated drug-induced repolarisation abnormalities using ToR-ORd population yielded similar results to the original study, with predicted risk being correct for 54 out of 62 compounds (87% accuracy). Compared to *Passini et al. (2017)*, the assessment of Mexiletine (a predominantly sodium blocker that is safe) was improved from false positive to true negative. High-dose Mexiletine led to formation of many EADs in ORd, but not in ToR-ORd (*Figure 7B*), highlighting the importance of the advances on sodium blockers presented in this work. At the same time, Procainamide and Metrodinazole were misclassified as false negatives compared to *Passini et al. (2017)*. However, these drugs are controversial, as Metrodinazole is considered non-torsadogenic by *Lancaster and Sobie (2016)*, and this study predicted both the drugs to be non-risky. Torsadogenic risk for all evaluated compounds and the confusion matrix of the classification are given in *Figure 7C*.

## Validation: response to disease

### Hyperkalemia

Hyperkalemia, the elevation of extracellular potassium, is a hallmark of acute myocardial ischemia caused by the occlusion of coronary artery. It was shown that hyperkalemia can significantly inhibit sodium channel excitability following repolarisation, leading to the prolongation of postrepolarisation refractoriness (*Coronel et al., 2012*). The dispersion of effective refractory periods (ERPs) between normal and ischemic zones forms a substrate for the initiation of re-entrant arrhythmia. In this new model, we tested the effect of hyperkalemia on tissue excitability using 1D fibres. As shown in *Figure 8A*, the elevation of extracellular potassium level led to an increase of the resting membrane potential (RMP) and the decrease of AP upstroke amplitude. As a result of weaker upstroke and more depolarised RMP, the APD shortened under hyperkalemia; however, the ERPs were prolonged due to the stronger sodium channel inactivation caused by the elevation of RMP (*Figure 8B*). Therefore, this new model successfully reproduced the longer post-repolarization refractoriness under hyperkalemia observed in experiments, and it can be used in the simulations of re-entrant arrhythmia under acute ischemia. In this regard, it presents an improvement over the original ORd model, which did not manifest postrepolarisation refractoriness without further modifications (*Dutta et al., 2017b*).

### Hypertrophic cardiomyopathy

Hypertrophic cardiomyopathy (HCM) is among the most common cardiomyopathies, manifesting as abnormal thickening of the cardiac muscle without an obvious cause (*Coppini et al., 2013*). Beyond

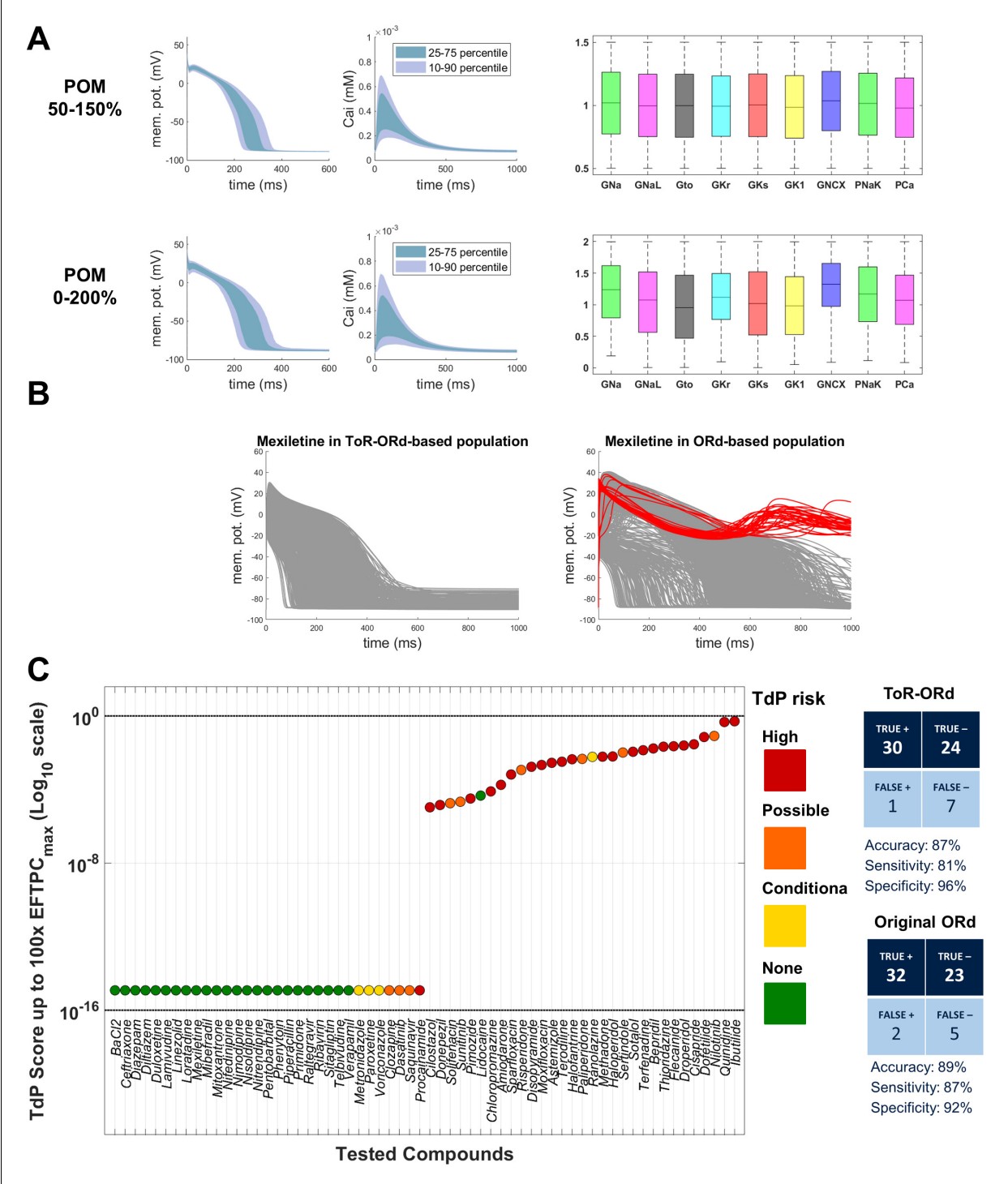

**Figure 7.** Populations of models and drug safety prediction. (**A**) Percentile-based summary of AP and calcium transient traces for the two populations of human ToR-ORd models (left side) and distribution of ionic current conductances among models in the population (right side), in the ranges [50-150]% (top row) and [0–200]% (bottom row) of the baseline values. (**B**) A comparison of 0–200% populations based on ToR-ORd (left) and ORd (right, based on **Passini et al., 2017**) in response to high-dose mexiletine (100-fold effective therapeutic dose). Traces classified as EADs are plotted in red (manifesting only in the ORd population). (**C**) TdP score obtained for simulations of the 62 reference compounds, based on the occurrence of drug-induced repolarisation abnormalities at all tested concentrations in the [0–200]% population of ToR-ORd models. The colours associated with drugs signify their established torsadogenicity as specified in Appendix 1-15.1.4. The logarithmic scale was considered to maximise the visual separation between safe and risky drugs. The classification based on the TdP score is summarised as a confusion matrix on the right, and also compared with the corresponding results obtained in a population of models based on the original ORd model (**Passini et al., 2017**).

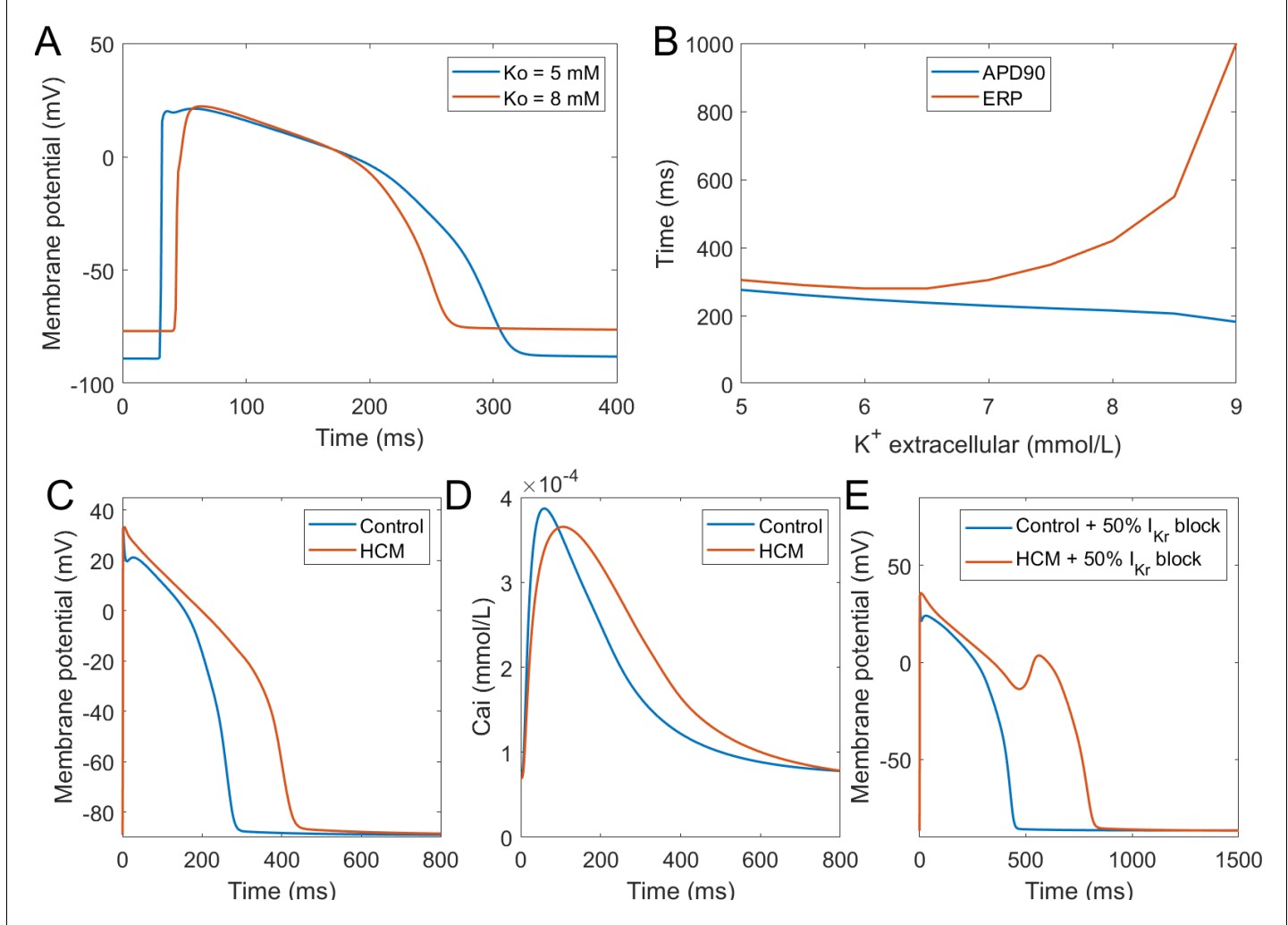

**Figure 8.** Simulation of hypertrophic cardiomyopathy (HCM) and hyperkalemia. (**A**) The effect of hyperkalemia on AP morphology; measured in the centre of a simulated fibre. (**B**) APD90 and effective refractory period (ERP) at varying extracellular potassium concentration. For extracellular potassium higher than 9 mM, full AP did not develop, but low-amplitude activation propagated through the fibre (Appendix 1-14). Membrane potential (**C**) and calcium transient (**D**) at 1 Hz pacing compared between a single healthy and HCM cell. (**E**) 50% I$_{Kr}$ block induces EADs in HCM cell, but not in a healthy one.

mechanical remodelling, the disease predisposes the hearts to arrhythmia formation, increasing the vulnerability to early afterdepolarisations. HCM induces complex multifactorial remodelling of cell electrophysiology and calcium handling, making it a challenging validation problem for a computer model. We applied the available human experimental data on HCM remodelling (based predominantly on *Coppini et al., 2013*) to our baseline model using an approach similar to *Passini et al. (2016)*, observing that the dominant features of the remodelling observed by Coppini et al. are captured. The HCM variant of the computer model corresponds to experimental data in the AP morphology, manifesting a significantly higher plateau potential and an overall APD prolongation (*Figure 8C*). The calcium transient amplitude of the HCM model is slightly reduced, has longer time to peak, and a noticeably longer duration at 90% recovery (*Figure 8D*), also consistent with the data by *Coppini et al. (2013)*. Ultimately, the HCM variant of our model is more prone to the formation of EADs (*Figure 8E*), as was shown experimentally (*Coppini et al., 2013*). This difference is in line with postulated key role of I$_{CaL}$ and NCX in EAD formation (*Luo and Rudy, 1994*; *Weiss et al., 2010*), both of which are markedly increased in HCM. Excessive prolongation of APD due to a strong increase in late sodium current in HCM also contributes to the EAD formation as well, as shown by *Coppini et al. (2013)*.

## Validation: human whole-ventricular simulations - from ionic currents to ECG

We conducted 3D electrophysiological simulations using the ToR-ORd model, representing the membrane kinetics of endocardial, epicardial and mid-myocardial cells to investigate their ability to simulate the ECG (see Appendix 1-15.1.5). Transmural and apex-to-base spatial heterogeneities as well as fibre orientations based on the Streeter rule were incorporated into a human ventricular anatomical model derived from cardiac magnetic resonance (*Lyon et al., 2018*).

*Figure 9A* shows the resulting electrocardiogram computed based on virtual electrodes positioned on a torso model shown in *Figure 9B*. The ECG manifests a QRS duration of 80 ms (normal range 78 ± 8 ms), and a QT interval of 350 ms (healthy:<430 ms); all of these quantitative measurements are in the range of ECGs of healthy persons (*Engblom et al., 2005*; *van Oosterom et al., 2000*). ECG morphology also showed normal features, such as R wave progression in the precordial leads from V1 to V6, isoelectric ST segment, and upright T waves in leads V2 to V6, with inverted T wave in aVR. *Figure 9C* shows the activation sequence is in agreement with *Durrer et al. (1970)*. The APD map shows longer APD in the endocardium and the base, and shorter APDs in the epicardium and the apex, respectively (*Figure 9D*).

## Discussion

In this study, we present a new model of human ventricular electrophysiology and excitation contraction coupling, which is able to replicate key features of human ventricular depolarisation, repolarisation and calcium transient dynamics. The ToR-ORd model was developed using a defined set of calibration criteria and subsequently validated on features not considered during calibration to demonstrate its predictive power. This article also unravels several important theoretical findings with implications for computational electrophysiology reaching beyond the ToR-ORd model and cardiac electrophysiology: firstly, the reformulation of the L-type calcium current, which is broadly relevant and generally applicable to human and other species, and secondly, the mechanistically guided replacement of $I_{Kr}$. Discovering the necessity to carry out these theoretical reformulations was enabled by the comprehensive set of calibration criteria and the use of a genetic algorithm to fulfil them. Finally, to enable reproducibility, we openly provide an automated model evaluation pipeline, which provides a rapid assessment of a comprehensive set of calibration or validation criteria.

The AP morphology of ToR-ORd is in agreement with the Szeged endocardial myocyte dataset used to construct the state-of-the art ORd model (*O'Hara et al., 2011*). The agreement is considerably better than that of ORd itself, which has important implications for multiple aspects studied in this work. The calcium transient also recapitulates key features of human myocyte measurements (*Coppini et al., 2013*). The validation of ToR-ORd shows that the model responds well to drug block with regards to APD (*Dutta et al., 2017a*). Good APD accommodation (reaction to abrupt, but persisting changes in pacing frequency) indicates a good balance between ionic currents (*Franz et al., 1988*; *Pueyo et al., 2010*). Replication of arrhythmia precursors such as early afterdepolarisations (*Guo et al., 2011*) and alternans (*Koller et al., 2005*) makes the model useful for simulations and understanding of arrhythmogenesis. This is particularly important in the context of heart disease, where ToR-ORd is shown to replicate key features of hyperkalemia (*Coronel et al., 2012*) and hypertrophic cardiomyopathy (*Coppini et al., 2013*). The model is also shown to be promising in drug safety testing, and whole-heart simulations demonstrate physiological conduction velocity (*Taggart et al., 2000*) and produce a plausible ECG signal. Among the improved behaviours compared to the state-of-the-art ORd model (*O'Hara et al., 2011*), the good response of the ToR-ORd model to sodium blockade is particularly noteworthy. ToR-ORd predicts the negative inotropic effect of sodium blockade, consistent with data (*Gottlieb et al., 1990*; *Tucker et al., 1982*; *Legrand et al., 1983*; *Bhattacharyya and Vassalle, 1982*), unlike ORd, which suggests a strong pro-inotropic effect. The improvement in ToR-ORd follows from the relatively complex interplay of the theoretically driven reformulation of the L-type calcium current and data-driven changes to the AP morphology. This result is of great importance in the context of pharmacological sodium blockers, but it also plays a crucial role in disease modelling, where both fast (*Pu and Boyden, 1997*) and late (*Coppini et al., 2013*) sodium current are altered.

An important feature of a model is its predictive power, and validation of a model using data not employed in model calibration is a central aspect of model credibility (*Pathmanathan and Gray,*

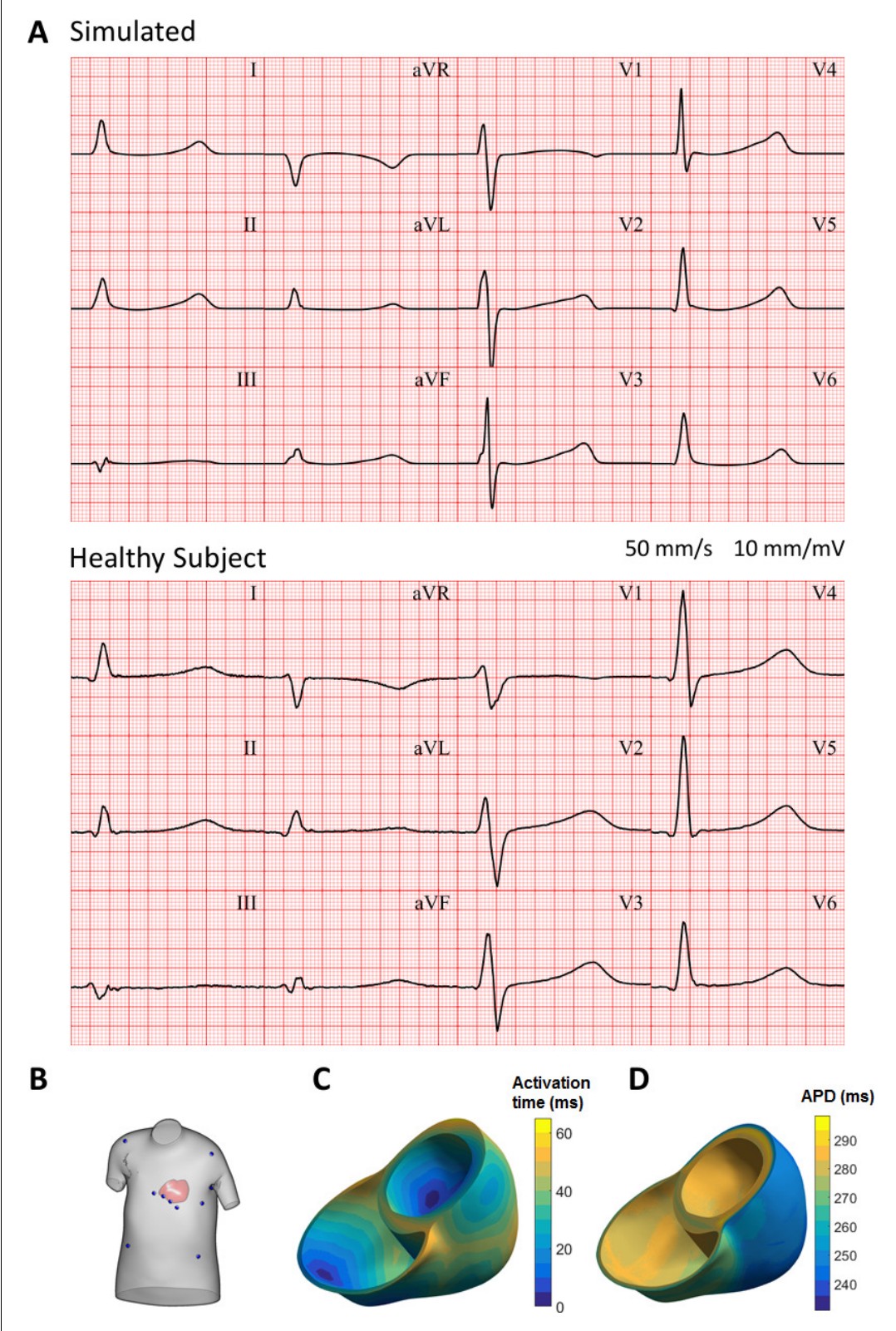

**Figure 9.** Simulated and clinical 12-lead electrocardiogram. (A) 12-lead ECGs at 1 Hz: simulation using the ToR-ORd model in an MRI-based human torso-ventricular model (top) and a healthy patient ECG record (bottom, https://physionet.org, PTB database, subject 122; *Bousseljot et al., 1995*; *Goldberger et al., 2000*). (B) Electrode positions on the simulated torso. (C) Activation time map. (D) APD map.

*2018*; *Carusi et al., 2012*). With this in mind, we designed our study to first calibrate the developed model using a set of given criteria, with subsequent validation of the model using separate data that were not optimised for during development. The fact that ToR-ORd manifests a wide range of behaviours consistent with experimental studies, even though it was not optimised for these purposes, suggests its generality and a large degree of credibility. To facilitate future model development, we also created an automated 'single-click' pipeline, which evaluates a wide range of calibration and validation criteria and creates a comprehensive HTML report. New follow-up models can thus be immediately tested against criteria presented here, making it clear which features of the model are improved and/or deteriorated by any changes made.

The greatest theoretical contribution of this work is the theory-driven reformulation of the L-type calcium current, namely the ionic activity coefficients and activation curve extraction. Activation curve of the current in previous cardiac models was based on the use of Nernst driving force in experimental studies, but the models then used Goldman-Hodgkin-Katz driving force to compute the current. This yields a theoretical inconsistency present in existing influential models of guinea pig, rabbit, dog, or human, for example (*Luo and Rudy, 1994*; *Hund et al., 2008*; *O'Hara et al., 2011*; *Shannon et al., 2004*; *Grandi et al., 2010*; *Carro et al., 2011*). We propose and demonstrate that in order to obtain consistent behaviour, the experimental I-V relationship measurements are to be normalised using the Goldman-Hodgkin-Katz driving force instead. Updated ionic activity coefficients and activation of the L-type calcium current improve key features of the current observed in the study underlying the ORd L-type calcium current model (*Magyar et al., 2000*), and strongly contribute to the improved reaction of the model to sodium blockade. The changes made are relevant in development of future models which use the Goldman-Hodgkin-Katz equation for L-type calcium current or other currents.

A second major contribution of this work reaching beyond the model itself is the set of observations on modelling of $I_{Kr}$, the dominant repolarising current in human ventricle. We noticed limitations of the ORd $I_{Kr}$ model, which may be a result of the single-pulse voltage clamp protocol to characterise the current behaviour. Approaches enabling the dissection of activation and recovery from inactivation based on more comprehensive experimental data, such as *Lu et al. (2001)* used in our work, may yield a more general and plausible model. In this study, this change was important predominantly for the response of the ventricular cell to calcium block, but our observations are highly relevant also for models of cells with naturally low plateau, such as Purkinje fibres or atrial myocytes.

We anticipate that the main future development of the presented model will focus on the ryanodine receptor and the respective release from sarcoplasmic reticulum. Similarly to most existing cardiac models, the equations governing the release depend directly on the L-type calcium current, rather than on the calcium concentration adjacent to the ryanodine receptors, which is the case in cardiomyocytes. Future development of the ryanodine receptor model and calcium handling will extend the applicability of the model to other calcium-driven modes of arrhythmogenesis, such as delayed afterdepolarisations. Also, while the model represents to a certain degree the locality of $I_{CaL}$ calcium influx and calcium release via the utilization of the junctional calcium subspace, a more direct representation of local control (*Stern, 1992*; *Hinch et al., 2004*), realistic spatially distributed calcium handling (*Colman et al., 2017*), or representation of stochasticity, may improve the insights the model can give into calcium-driven arrhythmogenesis. However, we note that such changes (particularly the detailed distributed calcium handling) will increase computational cost of the model's simulation. In addition, further research on the mechanisms regulating AP dependence on extracellular calcium concentration is needed to update this feature, not currently reproduced by most current human models (*Passini and Severi, 2014*).

## Acknowledgements

We are grateful to Prof. Yoram Rudy, Prof. Derek Terrar, and Dr. Derek Leishman for very useful discussions.

## Additional information

### Funding

| Funder | Grant reference number | Author |
|---|---|---|
| Wellcome | 100246/Z/12/Z | Blanca Rodriguez |
| Wellcome | 214290/Z/18/Z | Blanca Rodriguez |
| British Heart Foundation | FS/17/22/32644 | Alfonso Bueno-Orovio |
| European Commission | 675451 | Blanca Rodriguez |
| National Centre for the Replacement, Refinement and Reduction of Animals in Research | NC/P001076/1 | Blanca Rodriguez |
| European Federation of Pharmaceutical Industries and Associations | TransQST project (Innovative Medicines Initiative 2 Joint Undertaking 116030) | Blanca Rodriguez |
| BHF Centre of Research Excellence, Oxford | RE/13/1/30181 | Blanca Rodriguez |
| UK National Supercomputing | Archer RAP award (322 00180) | Blanca Rodriguez |
| Partnership for Advanced Computing in Europe AISBL | 2017174226 | Blanca Rodriguez |
| Amazon Web Services | Machine learning research award | Blanca Rodriguez |
| Horizon 2020 | TransQST project (Innovative Medicines Initiative 2 Joint Undertaking 116030) | Blanca Rodriguez |

The funders had no role in study design, data collection and interpretation, or the decision to submit the work for publication.

### Author contributions

Jakub Tomek, Conceptualization, Software, Formal analysis, Investigation, Visualization, Methodology, Project administration; Alfonso Bueno-Orovio, Resources, Software, Formal analysis, Validation, Investigation, Methodology; Elisa Passini, Ana Minchole, Formal analysis, Investigation, Visualization; Xin Zhou, Software, Formal analysis, Investigation, Visualization; Oliver Britton, Laszlo Virag, Andras Varro, Data curation; Chiara Bartolucci, Investigation; Stefano Severi, Supervision, Investigation; Alvin Shrier, Methodology; Blanca Rodriguez, Conceptualization, Resources, Supervision, Funding acquisition, Methodology, Project administration

### Author ORCIDs

Jakub Tomek (iD) https://orcid.org/0000-0002-0157-4386
Stefano Severi (iD) http://orcid.org/0000-0003-4306-8294

### Decision letter and Author response

Decision letter https://doi.org/10.7554/eLife.48890.sa1
Author response https://doi.org/10.7554/eLife.48890.sa2

## Additional files

### Data availability

No new experimental data were created. However, codes for simulations are available at https://github.com/jtmff/torord (copy archived at https://github.com/elifesciences-publications/torord).

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

## Appendix 1

# 1. Calibration criteria

This section provides additional information to the criteria listed in *Table 1*.

The AP morphology was based primarily on the large experimental dataset of human undiseased endocardial recordings from the Varró lab, published in *Britton et al. (2017)*. The ORd model (*O'Hara et al., 2011*) was based on a subset of these recordings. We aimed for similarity with the median of the AP data during the plateau and repolarization phase (from 15 ms after the AP peak). Two other datasets were used to confirm that early plateau potentials are ca. 20 mV, rather than the >30 mV as in ORd (*Coppini et al., 2013*; *Jost et al., 2013*).

The calcium transient morphology (CaT amplitude and duration at 90% recovery) was based on *Coppini et al. (2013)*, particularly given it is clear that the AP morphology is similar in their experimental recordings and the simulations with the TOR-ORd model. The aim was for the two CaT properties to lie within standard deviation of mean. A correction for the difference in APD with regards to CaT amplitude was made in Appendix 1-8.

The properties of $I_{CaL}$, the I-V relationship and steady-state inactivation were taken as reported in *Magyar et al. (2000)*, as this is the primary dataset used in the ORd $I_{CaL}$ construction. Visual assessment of simulations versus data was used.

Negative inotropy of sodium blockers was based on *Gottlieb et al. (1990)*; *Tucker et al. (1982)*; *Legrand et al. (1983)*; *Bhattacharyya and Vassalle (1982)*, which report 8–28% reduction in whole-heart contractility, depending on drug, dose and index of contractility. Given the variability, throughout the calibration of a single-cell model, we aimed for any reduction in CaT amplitude following 50% reduction of $I_{Na}$ and $I_{NaL}$.

The blockade of $I_{CaL}$ is known to shorten APD across species, including human (*O'Hara et al., 2011*). Within the process of calibration, we aimed for any APD shortening at 50% $I_{CaL}$ reduction.

EADs were shown to form under ca. 85% $I_{Kr}$ block in human myocytes at 0.25 Hz pacing (*Guo et al., 2011*). Thus, we aimed for the new model to manifest EADs of similar amplitude as in the data (ca. 14 mV) in corresponding conditions.

APD alternans was observed in undiseased human cells at rapid pacing (*Koller et al., 2005*). We aimed for a model manifesting APD alternans, with the onset at basic cycle length shorter than 300 ms.

The reported conduction velocity in human heart is 65 m/s (*Taggart et al., 2000*), and we compared this value to the result of a fibre simulation using the developed ToR-ORd model.

# 2. Genetic algorithm fitness

The fitness function has 18 inputs, 16 of which are the multipliers of conductances for the following currents and fluxes: $I_{Na}$, $I_{CaL}$, $I_{to}$, $I_{NaL}$, $I_{Kr}$, $I_{Ks}$, $I_{K1}$, $I_{Kb}$, $I_{NaCa}$, $I_{NaK}$, $I_{Nab}$, $I_{Cab}$, $I_{pCa}$, $I_{CaCl}$, $J_{rel}$, $J_{up}$. Also varied were $K_{\infty,Rel}$, a parameter of $J_{rel}$ (constrained between 0.9 and 1.7) and the fraction of $I_{NaCa}$ in the junctional subspace (constrained between 0.18 and 0.4).

If the genetic algorithm (GA) employed symmetric percentual variations of raw current multipliers (e.g., + /- 50%), it would not sample the input space evenly – for example, a symmetric Gaussian mutation is much more likely to halve a parameter than to double it (assuming the Gaussian curve being centred at 1, the density in 0.5 is the same as in 1.5, but not in 2), while the likelihood should be arguably the same. The initial population would also likely be highly skewed towards current reduction. In order to avoid these issues, we made the GA work internally with logarithms of multipliers, which makes the sampling symmetrical (-log (0.5)=log(2), etc.). The fitness function first exponentiates the log-multipliers to obtain the actual multipliers, and these are subsequently used in further simulation.

Within the fitness function, the evolved model is pre-paced for 130 beats at 1 Hz, with the final state $X_{130}$. Subsequently, 20 more beats are simulated at the following conditions, with $X_{130}$ as the starting state: a) no change to parameters, b) sodium blockade (50% $I_{Na}$ block, 50% $I_{NaL}$ block), c) calcium blockade (50% $I_{CaL}$ block), d) $I_{K1}$ blockade (50% block). 150 beats in

total for the control condition is a compromise between total runtime and similarity to the stable-state behaviour. The 20 beats at various conditions following 130 beats of prepacing are sufficient to manifest the effects of the respective blocks , while keeping the runtime low (When the model is evaluated in the manuscript, the outputs are based on 1000 beats of pre-pacing; the 150 or 130+20 beats are used only during model refitting to allow sufficiently fast runtime.).

Based on these simulations, a two-element fitness vector is obtained; the first element describes similarity of action potential (AP) morphology to the reference, with the second element aggregating other criteria (calcium transient duration and amplitude, calcium transient amplitude reduction with sodium blockade, action potential duration (APD) reduction with calcium blockade, and a depolarisation with $I_{K1}$ block).

Not all calibration criteria (**Table 1**) were represented in the fitness function, as simulating all corresponding protocols would be prohibitively slow. Instead, the calibration criteria not optimized using the genetic algorithm were subsequently fulfilled by mechanistically informed manual changes, while making sure the already optimised criteria were not violated.

## 3. Extraction of $I_{CaL}$ activation from I-V relationship

The experimental protocol to measure I-V relationship of the $I_{CaL}$ uses square pulse stimuli to measure peak current for different pulse potentials (**Appendix 1—figure 1**, top left). The peak current can be seen as the product of two components: activation (how open the channels are) and driving force (the strength of the diffusion and electrical gradients that produce ionic flow through the open channels). To obtain the activation value for each pulse potential, the corresponding peak current is divided by the theoretical estimate of the driving force. The resulting curve can then be scaled to 0–1, producing estimated fractional activation (**Appendix 1—figure 1**, top right). The driving force may be computed based on the linear equation V-$E_{Ca}$, or the nonlinear Goldman-Hodgkin-Katz (GHK) flux equation (**Appendix 1— figure 1**, bottom left and bottom right, respectively). This yields the two corresponding activation curves in **Appendix 1—figure 1**, top right.

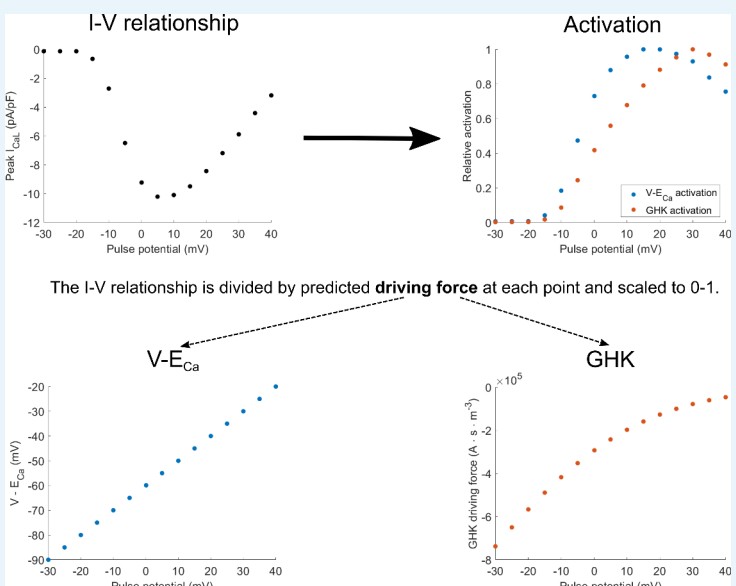

**Appendix 1—figure 1.** Diagram of how activation curves are extracted from the I-V relationship. The activation curves (top right) are obtained by dividing values in the I-V curve (top left) at each pulse potential by the driving force considering either V-$E_{Ca}$-based driving force (bottom right) or the Goldman-Hodgkin-Katz (GHK) flux equation (bottom left).

**Appendix 1—figure 1** can be also used to illustrate the theoretical inconsistency in the ORd model and many other models (e.g. **Luo and Rudy, 1994**; **Hund et al., 2008**;

*O'Hara et al., 2011*; *Shannon et al., 2004*; *Grandi et al., 2010*; *Carro et al., 2011*): the V-E$_{Ca}$ driving force is used to obtain the activation curve, but then the I$_{CaL}$ is computed using the GHK driving force, which does not yield the original data, as illustrated in the Results section. Returning to the notion of I-V relationship as a product of activation and driving force, the same shape of I-V relationship can be obtained by pointwise multiplication of the blue driving force (bottom, left) with blue activation (top, right), or the red driving force (bottom, right) with red activation (top, right). However, simulating the I-V relationship using the ORd model corresponds to the product of blue activation and the red driving force, which then naturally does not match greatly the original I-V relationship (*Figure 2D*, or *Appendix 1—figure 2* in Appendix 1-4).

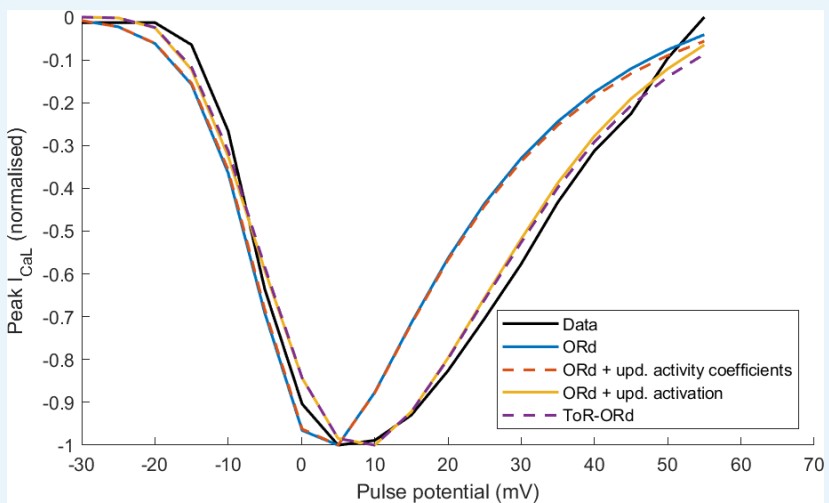

**Appendix 1—figure 2.** Effect of the proposed changes in the activation curve and activation coefficient on the I-V relationship of I$_{CaL}$ in ORd versus ToR-ORd.

## 4.I$_{CaL}$ updates and the I-V relationship

*Appendix 1—figure 2* illustrates the effect that updates in activation curve and activation coefficients have on the simulated I$_{CaL}$ I-V relationship considering four models:

- Control ORd
- ORd with updated activity coefficients as in ToR-ORd (fixed to 0.6532/0.6117 for intracellular/extracellular calcium, 0.89/0.88 for intracellular/extracellular sodium; the same also for potassium)
- ORd with updated activation curve as in ToR-ORd • ToR-ORd model.

The I-V curves were normalised to a minimum of −1, in order to focus on the shape of the I-V curve rather than its amplitude (which is modulated by maximal conductance). Considering the new activation curve extracted using the GHK equation in the ORd model yields a near-identical I-V curve to the ToR-ORd model (*Appendix 1—figure 2*). This demonstrates this is the key mechanism in the I-V curve improvement. The update of activity coefficients did not alter the shape of the I-V relationship noticeably compared to the ORd model (*Appendix 1—figure 2*), but played a key role in avoiding the reversal of I$_{CaL}$ illustrated in *Figure 2F* and a good response to sodium block, as explained in the next section.

## 5. Extraction of steady-state inactivation of I$_{CaL}$

As shown in *Figure 2E* of the main manuscript, the steady-state inactivation curve of the ORd model differs from the experimental data. This is surprising, given that both the steady state voltage and calcium inactivation in the model are a direct fit to the experimental data. The explanation of this phenomenon lies in the construction of the I$_{CaL}$ equation in the ORd model,

which is as follows (only the non-phosphorylated part is given; the phosphorylated is analogous):

$$I_{CaL,\,NP} = \bar{I}_{CaL} \cdot d \cdot (f \cdot (1-n) + f_{Ca} \cdot n \cdot j_{Ca})$$

where $\bar{I}_{CaL}$ is the product of driving force and channel conductance, $d$ is activation, $f$ is voltage-driven inactivation, $n$ is a weight of calcium inactivation, $f_{Ca}$ is calcium-driven inactivation, and $j_{Ca}$ corresponds to recovery from inactivation. In the ORd model, stable-state values of $f, f_{Ca}, j_{Ca}$ are all fit to the steady-state inactivation in **Magyar et al. (2000)**. During a long voltage clamp pulse used to inactivate $I_{CaL}$, $\bar{I}_{CaL}$ and $d$ can be assumed to be constant, leaving the remaining parentheses as the factor determining measured inactivation. If $j_{Ca}$ was equal to 1, then the equation would reduce to the experimentally observed inactivation in the steady state (however, this prevents formation of EADs). However, given that $j_{Ca}$ effectively acts as a second inactivation, the $j_{Ca} \cdot f_{Ca}$ product corresponds to a square of the experimentally observed inactivation, making the inactivation stronger. Thus, for $n > 0$, this $I_{CaL}$ model manifests total steady-state inactivation stronger than the one from **Magyar et al. (2000)**.

As shown in **Figure 2E** of the main manuscript, this problem is not present in the ToR-ORd model, which yields a better match with experimental data than ORd. Inspecting state variables at the inactivating pulse potentials where the models differ (approximately between -30 and -5 mV), we observed that the ORd model has a higher level of calcium in the subspace, increasing the value of the $n$ variable, making the contribution of the overestimated inactivation stronger. The higher calcium level in the subspace in the ORd model seems to arise from a combination of two factors: (1) Overestimated activation and thus enhanced calcium entry in ORd (evident from **Figure 2 C, D** in the main manuscript), accompanied by small differences in $J_{rel}$ formulation translating into more SR release in ORd; (2) less total NCX in the subspace, corresponding to less calcium clearance.

It is worth noting that the exact shape of the simulation-based inactivation curve depends on how exactly the conditions in **Magyar et al. (2000)** are represented. As explained in Appendix 1-15.1.1, intracellular and potassium are fixed when the steady-state inactivation is measured, but calcium is not fixed. If the intracellular calcium in the model was also clamped to the zero concentration of the pipette, $n$ would be zero and both ORd and ToR-ORd would fit the inactivation curve well.

## 6. $I_{CaL}$ reversal and ionic activity coefficients

As illustrated in **Figure 2F**, the ToR-ORd model yields a consistently inward $I_{CaL}$ during the AP, whereas for the ORd, there is a reversal to outward $I_{CaL}$ following the peak of the AP. For the ORd midmyocardial cell, $I_{CaL}$ is highly positive ($I_{CaL}$ = 3.32 pA/pF) at t = 4.3 ms. This can be explained by the update in the ionic activity coefficients described in section *Methods: Determining ionic activity coefficients*. The sign of $I_{CaL}$ is determined by the sign of the driving force (other components of $I_{CaL}$ are the permeability constant, and gating variables, both of which are nonnegative). The GHK equation for driving force (with all elements explained in Materials and methods of the main manuscript) can be divided into four components for clarity (**Appendix 1—figure 3**). Components 1 and 4 are positive (V = 33 mV at this point of simulation) and thus do not affect the sign; therefore, the sign of the driving force in this case is determined by the relative size of components 2 and 3. In control ORd (with activity coefficients of 1 and 0.341 for intracellular/extracellular space), we have, for components C2 and C3:

$$C2 = [S]_i \cdot e^{\frac{z V \cdot F}{R T}} = \gamma_i \cdot m_i \cdot e^{\frac{z V \cdot F}{R T}} = 1 \cdot 0.0822 \cdot 11.6794 = 0.9598$$

$$C3 = [S]_o = \gamma_o \cdot m_o = 0.341 \cdot 1.8 = 0.6138$$

where $\gamma_i$, $\gamma_o$ are intra/extracellular activity coefficients, $m_i$ is the calcium concentration in

junctional subspace, and $m_o$ is extracellular calcium concentration. Therefore, C2 > C3 and the sign of driving force (and thus also $I_{CaL}$) is positive.

$$\varphi_{CaL} = z^2 \cdot \frac{V \cdot F^2}{R \cdot T} \cdot \frac{[S]_i \cdot e^{\frac{z \cdot V \cdot F}{R \cdot T}} - [S]_o}{e^{\frac{z \cdot V \cdot F}{R \cdot T}} - 1},$$

**Appendix 1—figure 3.** Goldman-Hodgkin-Katz equation with colour coding of components.

However, if we use ionic activity coefficients based on ToR-ORd (0.6532 and 0.6117 for intracellular/extracellular), we get:

$$C2 = \gamma_i \cdot m_i \cdot e^{\frac{z \cdot V \cdot F}{R \cdot T}} = 0.6532 \cdot 0.0822 \cdot 11.6794 = 0.6269$$

$$C3 = \gamma_o \cdot m_o = 0.6117 \cdot 1.8 = 1.1011$$

Therefore, in this case, C2 < C3, and the sign of driving force and $I_{CaL}$ is negative and thus consistently inward during the AP.

## 7. P2/P1 protocol for $I_{CaL}$

We compared the results of the ToR-ORd and ORd models to experimental data in human myocytes for the P2/P1 protocol reported by *Fülöp et al. (2004)* as previously shown in the original ORd model study (*O'Hara et al., 2011*). In this protocol, two rectangular pulses (from −40 mV to 10 mV, lasting either 25 ms or 100 ms) are applied, and the ratio of peak $I_{CaL}$ during the second pulse versus the first one is computed. Simulated curves with both models are consistent with the data in that a) 25 ms pulse curve lies above the 100 ms pulse curve, b) shorter coupling interval is associated with less current availability (*Appendix 1—figure 4*). Both models show an offset to the experimental data. This could be due to a) Difference in $I_{CaL}$ behaviour/density and/or calcium loading (affecting inactivation) between the Fülöp (P2/P1 protocol) and Magyar (the main basis for the $I_{CaL}$ model) studies. It could be also linked to mechanisms of $I_{CaL}$ inactivation not represented in the model (e.g. the inactivation by the calcium flow through the channel, represented for example by *Mahajan et al., 2008*).

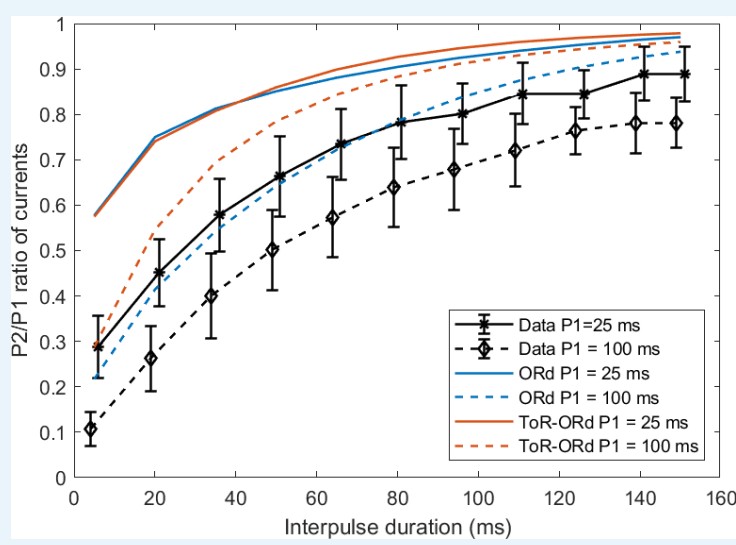

**Appendix 1—figure 4.** P2/P1 protocol for two pulse durations.

## 8. Calcium transient amplitude in ToR-ORd

The calcium transient amplitude of 312 nM in baseline ToR-ORd is slightly below the mean ± std range of 350 (330-370) nM in *Coppini et al. (2013)*. However, we hypothesised that the difference might be due to the different APD, which affects the calcium loading of the cell by modulating $I_{CaL}$ duration. When the APD of the ToR-ORd was extended to 351.5 ms (close to ca. 350 ms reported by Coppini et al.) by reducing $I_{Kr}$ by 45%, the calcium transient amplitude increased to 360 nM, which is close to the data mean. After such an AP prolongation, both time to peak (54 ms) and calcium transient duration at 90% recovery (459 ms) remained within standard deviation of the data.

## 9. Summary of literature on sodium blockers, $I_{Na}$, and $I_{NaL}$ reduction

The exact ratio of pharmacological block of $I_{Na}$ and $I_{NaL}$ is drug and dose-dependent, with late sodium current generally being blocked somewhat more than the fast sodium current. For flecainide applied to wild-type $Na_v1.5$, the IC50 value of $I_{Na}$ and $I_{NaL}$ were 127 ± 6 and 44 ± 2 µM, respectively; for example, for 50% $I_{Na}$ block, a 75% $I_{NaL}$ block would be expected (*Nagatomo et al., 2000*). In the case of TTX, the IC50 values appear to be more similar for the two currents: 1.2 µM for $I_{Na}$ (*Bradley et al., 2013*) and 0.95 µM for $I_{NaL}$ (*Horvath et al., 2013*). Lidocaine also appears to block $I_{NaL}$ preferentially: the measured IC50 for $I_{NaL}$ is 10.79 µM (*Crumb et al., 2016*), which is lower than 44 µM measured for $I_{Na}$ (Janssen Pharmaceutical internal database, referred to in *Passini et al., 2017*). However, we note that comparing IC50 values between studies has limited quantitative relevance, given expected differences between conditions and protocols.

## 10. Role of $I_{CaL}$ properties and AP morphology in calcium transient changes by sodium blockade

The changes in AP morphology and $I_{CaL}$ introduced in the ToR-ORd model improved its response to Na currents block, particularly with respect to reduction of the Ca transient amplitude (illustrated in *Figure 3*). To further dissect the contribution of differences between ToR-ORd and ORd in AP morphology and $I_{CaL}$ properties, we simulated two AP clamp scenarios using the four different models considered previously in *Appendix 1—figure 2* (i.e. ORd, ORd and ToR-ORd activity coefficients, ORd and ToR-ORd $I_{CaL}$ activation curve, and

ToR-ORd). The first scenario applied an AP clamp based on ORd AP morphologies for control and for 50/50% $I_{Na}/I_{NaL}$ block (as shown in *Figure 3B*). The second scenario considered AP clamps based on ToR-ORd AP morphologies in control and 50/50% $I_{Na}/I_{NaL}$ block (as shown in *Figure 3A*). For each of the four models and the two AP clamp scenarios, the ratio of calcium transient amplitudes (sodium-block versus control) was computed (*Appendix 1—table 1*). This allowed quantifying the importance of differences in AP morphology and updates to the $I_{CaL}$ representation with regard to CaT amplitude changes under sodium block.

**Appendix 1—table 1.** Dissection of improvement in reaction to sodium blockade. Ratios of calcium transient amplitudes for AP clamps corresponding to baseline and sodium-blocked models, arising from ORd and ToR-ORd. Row/column numbering used in the text does not include the header and the first column which describes the models (i.e. the text refers to four rows and two numerical columns).

| Model | ORd AP clamps | ToR-ORd AP clamps |
|---|---|---|
| M1 (ORd) | 1.366 | 1.061 |
| M2 (ORd with ToR-ORd activity coefficients) | 1.301 | 0.999 |
| M3 (ORd with ToR-ORd activation curve) | 1.326 | 1.015 |
| M4 (ToR-ORd) | 1.193 | 0.985 |

A value below one in *Appendix 1—table 1* indicates the expected decrease in calcium transient amplitude with sodium block compared to control. This was achieved only by using the ToR-ORd AP clamp with the ToR-ORd model (row 4, column 2), and also but to a lesser extent with the ToR-ORd AP clamp and the ORd with ToR-ORd activity coefficients (row 2, column 2). Considering the ToR-ORd AP morphology (versus $I_{CaL}$ activation coefficients or activation curves) has the strongest effect in changes in calcium transient amplitude with sodium blockade. This is demonstrated by the universally lower values in the second column of *Appendix 1—table 1* compared to the first column. The update of ionic activity coefficients alone leads to a reduced increase in CaT amplitude (in ORd), or a reduction in CaT amplitude (in ToR-ORd) following sodium blockade (*Appendix 1—table 1*, row 2 versus 1). Updated activation curve leads to a less pronounced increase in calcium transient amplitude in both models (*Appendix 1—table 1*, row 3 versus 1). Using the fully updated ToR-ORd model for simulations gave the most pronounced reduction in how much the calcium transient is increased upon sodium blockade using ORd AP clamps (*Appendix 1—table 1*, row 4 versus 1; column 1). It also yielded the most pronounced reduction in calcium transient amplitude following sodium blockade when using ToR-ORd AP clamps (*Appendix 1—table 1*, row 4 versus 1; column 2).

AP morphology and peak AP levels regulate calcium transient amplitude through $I_{CaL}$, and specifically through its voltage-dependency of activation curve, driving force and inactivation. It is important to note key differences between ToR-ORd and ORd models in this respect. In the ToR-ORd model, the AP morphology and the $I_{CaL}$ activation curve (*Figure 2C*) mean that $I_{Na}$ blockade leads to a reduction of peak $I_{CaL}$ activation. Conversely, in ORd, the activation curve is flat from 15 mV on (*Figure 2C*), and activation is not affected when sodium block reduces peak and early-plateau potential from 40 mV to 30–35 mV. At the same time, lowered peak and early-plateau potentials following sodium blockade increase $I_{CaL}$ driving force and weaken the voltage-driven inactivation, enhancing total $I_{CaL}$. Furthermore, in the ToR-ORd model, AP with and without sodium block reach a near-identical notch and early-plateau potential soon after the peak (*Figure 3A*). However, in the ORd, the difference in early-plateau membrane potentials lasts almost 30 ms after the peak (*Figure 3B*), prolonging the effect of increased driving force and reduced inactivation on $I_{CaL}$, which is furthermore not compensated by reduction in activation.

To further explore the impact of AP morphology on $I_{CaL}$ properties, $I_{CaL}$ activation and driving force was computed with the ToR-ORd model, again under four AP clamps considering AP morphologies shown in *Figure 3A,B* (obtained with ToR-ORd control and 50/50% $I_{Na}/I_{NaL}$ block, and ORd control and 50/50% $I_{Na}/I_{NaL}$ block). The results are presented in *Appendix 1—figure 5*. The activation with ToR-ORd AP clamp is reduced with sodium block

versus control AP clamp morphologies (*Appendix 1—figure 5A*, yellow vs purple), given the difference in peak membrane potentials (*Figure 3A*) and their position on the activation curve (*Figure 2C*). However, the activation over time is similar with ORd AP morphology in control versus sodium block (*Appendix 1—figure 5A*, red vs blue), as the membrane potential is so high that full activation is reached in both cases.

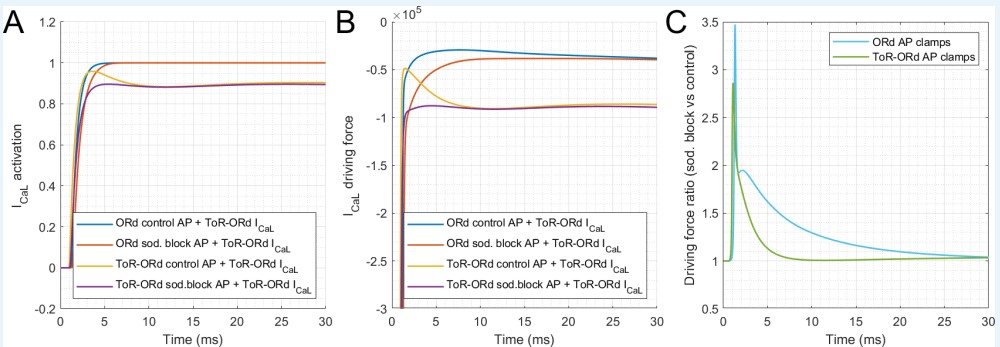

**Appendix 1—figure 5.** Effect of AP morphology on $I_{CaL}$ variables under sodium blockade. (**A**) $I_{CaL}$ activation. (**B**) ICaL driving force. In (**C**) are shown ratios of curves from (**B**) red pointwise-divided by blue (shown in light blue) and purple pointwise-divided by yellow (shown in green).

Furthermore, the $I_{CaL}$ driving force obtained with the four AP clamps is shown in *Appendix 1—figure 5B* (more negative values correspond to a greater driving force), and *Appendix 1—figure 5C* displays the ratio of the driving forces between sodium-blocked AP clamp and control AP clamp for ORd and ToR-ORd morphologies. It is clear that the driving force elevation under sodium-block AP morphology versus control is much smaller and shorter lasting when clamping to ToR-ORd than ORd AP morphologies (*Appendix 1—figure 5C*).

Therefore, the combination of ToR-ORd AP morphology and $I_{CaL}$ activation curve (which is consistent with experiments and with lower peak and early plateau voltages compared to ORd) result in reduction in both activation and driving force of $I_{CaL}$ with sodium bock. This explains how $I_{Na}$ block causes a smaller increase in $I_{CaL}$ and thus also in calcium transient amplitude using the ToR-ORd versus ORd models (*Figure 3E and F*). This smaller calcium increase due to $I_{Na}$ block is further compensated by the reduction in calcium entry and loading caused by $I_{NaL}$ block. The effect of $I_{NaL}$ on the simulated cell's calcium loading is mediated by the NCX (reduced sodium influx via $I_{NaL}$ reduces calcium influx via NCX) and by $I_{CaL}$ (APD shortening induced by $I_{NaL}$ reduction also shortens $I_{CaL}$, reducing calcium influx). Given that the effect of fixed-amount $I_{NaL}$ reduction on APD is stronger in ToR-ORd compared to ORd (*Figure 3A,B*), $I_{NaL}$ loss reduces calcium transient amplitude more in ToR-ORd.

## 11. Sodium block in fibre

In order to assess how cell coupling affects the sodium block behaviour, we simulated half-blocks of $I_{Na}$ and $I_{NaL}$ in fibre (*Appendix 1—figure 6*). The effect of the respective blocks is generally consistent with single-cell behaviour (*Figure 3*) in that $I_{Na}$ block increases calcium transient amplitude, and $I_{NaL}$ block reduces it. The effect of both half-blocks combined shows the same trend as the single-cell but is of greater magnitude: ToR-ORd shows a greater reduction in the calcium transient amplitude, while ORd shows a greater increase. In this section, a version of ORd with updated $I_{Na}$ to allow for good propagation was used (*Passini et al., 2016*), and the tissue conductivity was set to achieve the conduction velocity of 63 m/s.

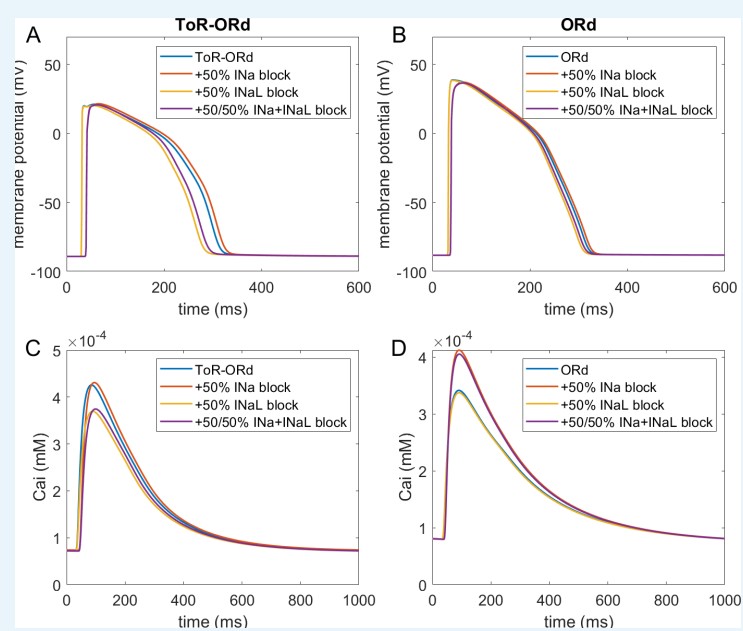

**Appendix 1—figure 6.** Effect of sodium blockers on ToR-ORd and ORd in fibre. Membrane potential in fibre at four simulated conditions (control, half block of fast sodium, half block of late sodium, and half block of both fast and late sodium currents) in ToR-ORd (**A**) and ORd (**B**). In (**C, D**) is given the corresponding calcium transient. The measurements were taken in the node at the half of the fibre; consequently, action potentials in the conditions including $I_{Na}$ block are delayed to the other traces, given corresponding conduction delay.

## 12. Calibration: $I_{CaL}$ block and $I_{Kr}$ replacement

During the model development (with the original ORd formulation of $I_{Kr}$ being used at the stage), we observed that in models with good AP morphology and response to sodium block, 50% $I_{CaL}$ block resulted in APD prolongation, which is in conflict with experimental evidence. Dissecting the cause of this phenomenon, we noticed considerably diminished $I_{Kr}$ density upon such treatment. Given that $I_{Kr}$ is the dominant repolarising current in human ventricle and human-specific computer models, this led to a net APD prolongation. The diminished $I_{Kr}$ density is due to the lack of activation of $I_{Kr}$ following from the combination of data-like AP morphology and formulation of fast and slow time constant of activation (Figure 3A in *O'Hara et al., 2011*). These time constants indicate a rapid activation between 25 and 40 mV, but when the membrane potential is reduced below 15 mV, activation time increases steeply (both fast and slow time constant are >1000 ms at 0 mV). Therefore, when the 50% $I_{CaL}$ block reduces plateau potential (a reduction of ca. 10 mV from control data-driven values of ca. 20 mV at 10 ms, 10 mV at 100 ms), $I_{Kr}$ activation is slowed greatly, producing a reduction in current density.

Subsequently, we sought to understand why this problem was not present in the original ORd model, which was shown to respond well to $I_{CaL}$ block with regard to APD (*O'Hara et al., 2011*; *Dutta et al., 2017a*). This follows from the high-plateau AP configuration of ORd (ca. 38 mV at 10 ms and 20 mV at 100 ms); when such a high plateau is lowered by 10 mV, it still falls within the zone of rapid $I_{Kr}$ activation. Under such conditions, the loss of $I_{CaL}$ shortens the APD much more than the minor loss of $I_{Kr}$ prolongs it, with the net effect being APD shortening.

Importantly, the article describing the Lu-Vandenberg model of $I_{Kr}$ (*Lu et al., 2001*) used in the final version of ToR- ORd also provides $I_{Kr}$ measurements under AP clamp. The study utilises AP clamps of both normal-plateau ventricular AP and low-plateau Purkinje AP (which can be taken as a proxy for $I_{CaL}$ blocked ventricular cell). The experimentally measured peak $I_{Kr}$ is slightly reduced in low-plateau AP clamp versus the high-plateau one (reduction to

70%). This is generally captured by simulations using our version of the Lu-Vandenberg model (reduction to 57%, *Appendix 1—figure 7*). However, peak $I_{Kr}$ in ORd reduces to mere 15% in the low-plateau AP clamp versus high-plateau one (*Appendix 1—figure 7*), which is due to the slow time constant of activation as described above. The fact that $I_{Kr}$ is capable of rapid activation (or recovery from inactivation, mimicking activation) is independently corroborated also by the elegant study of sinusoidal voltage pulse protocols by *Beattie et al. (2018)*, where $I_{Kr}$ responds noticeably to membrane potential changes around 0 mV.

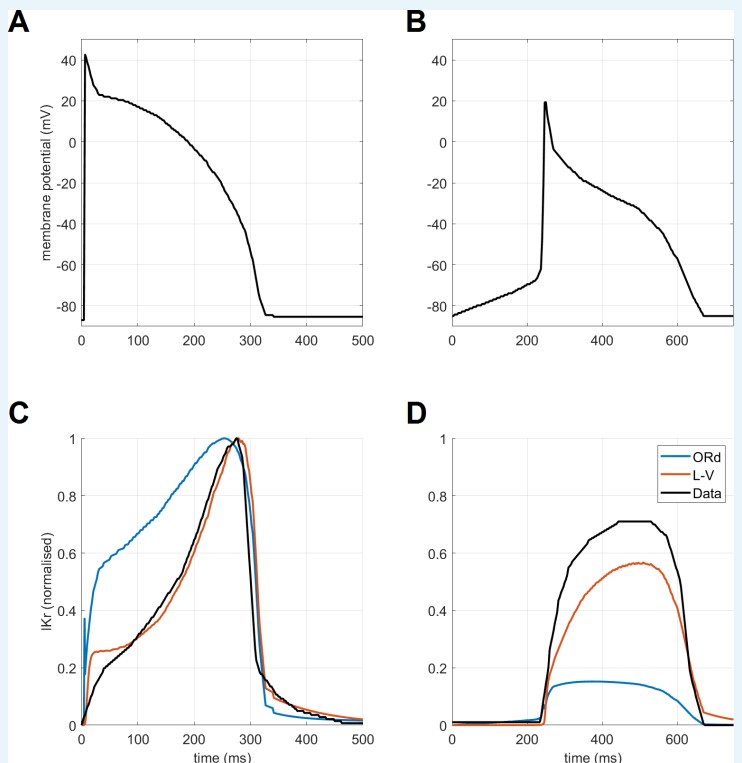

**Appendix 1—figure 7.** AP clamps and resulting $I_{Kr}$. (**A,B**) Endocardial and Purkinje AP used as an AP clamp (5th beat of the simulation shown). (**C**) $I_{Kr}$ produced by the endocardial AP. (**D**) $I_{Kr}$ produced by the Purkinje AP. Shown are data manually extracted from *Lu et al. (2001)*, as well as simulations using ORd and ToR-ORd (L-V, coding for Lu-Vandenberg hERG model). A normalisation was performed in (**C**), (**D**) to facilitate the relative loss of $I_{Kr}$ using the low-plateau AP shape. In (**C**), the three traces were normalised to 0–1 range via division by maxima of the traces: $max_{Data}$, $max_{ORd}$, $max_{LV}$. Subsequently, in (**D**), the data trace was divided by $max_{Data}$, ORd simulation by $max_{ORd}$, and the ToR-ORd model simulation by $max_{LV}$. Thus, both traces produced by a given model (or measured experimentally) are divided by the same number, allowing the assessment of relative reduction of $I_{Kr}$ at the low-plateau AP.

## 13. Validation: drug block and APD

In addition to four drugs and their effect on APD reported in the main manuscript *Dutta et al. (2017b)*, also studied the effect of $BaCl_2$, modelled as a 90% $IK_1$ block. However, we excluded considering this drug fundamentally due to the uncertainty on its multichannel effects on several potassium currents, in addition to $I_{K1}$. Specifically, it was previously shown that $BaCl_2$ also blocks $I_{Kr}$ and $I_{Ks}$ at the used concentration (*Weerapura et al., 2004*; *Gibor et al., 2004*). Furthermore, as shown in *Kristóf et al. (2012)*, even a lower dose of $BaCl_2$ prolongs the AP plateau and early phase of repolarization, thus at membrane potentials at which $IK_1$ is not activated. This corroborates

that $BaCl_2$ is likely to block $I_{Kr}$ and $I_{Ks}$, in addition to $I_{K1}$, but to an uncertain degree. Furthermore, pure block of $I_{K1}$ in the ORd model results in APD prolongation from around −50 mV only, which indicates a clear discrepancy with the experimental data.

To further explore this issue, we simulated the effect of pure 90% $I_{K1}$ block (which is consistent with the concentration of $BaCl_2$ used in the experiments) as well as additional $I_{Kr}$ block using the ToR-ORd. The results are shown in *Appendix 1—table 2*.

**Appendix 1—table 2.** Effect of $BaCl_2$ on APD90 in experiments and simulations using the ToR-ORd model with different degrees of $I_{K1}$ and $I_{Kr}$ block.

| Basic cycle length | Experimental data (*O'Hara et al., 2011*) | Simulated 90% $I_{K1}$ block | Simulated 90% $I_{K1}$ + 15% $I_{Kr}$ block | Simulated 90% $I_{K}1$ + 30% $I_{Kr}$ block |
|---|---|---|---|---|
| 500 ms | 271 (±50) ms | 250 ms | 271 ms | 297 ms |
| 1000 ms | 306 (±50) ms | 281 ms | 307 ms | 341 ms |
| 2000 ms | 361 (±50) ms | 286 ms | 311 ms | 342 ms |

For the three basic cycle length tested, the simulations are in overall agreement with the experimental data (*O'Hara et al., 2011*), with an improved match for 90% $I_{K1}$ + 15% $I_{Kr}$ block. It is however uncertain (albeit likely) whether the block of other currents, such as for example $I_{Ks}$ contributes to explain the effect of $BaCl_2$ on the APD.

A second matter discussed in this section is whether the assessment of nisoldipine in *Figure 5D* classifies as a calibration or validation result. With respect to $I_{CaL}$ block, the genetic algorithm optimisation contained a criterion promoting APD shortening in response to a 50% calcium blockade; however, the degree of block was different (90% when assessing nisoldipine), and the optimiser did not aim specifically for the APD reported by *Dutta et al. (2017a)*. Additionally, the $I_{Kr}$ formulation substitution was motivated by the poor performance of the original $I_{Kr}$ model in response to 50% $I_{CaL}$ blockade. After the substitution was performed and the $I_{Kr}$ conductance scaled to achieve similar APD at 1 Hz pacing at control conditions, no further optimization towards the experimental data in *Figure 5D* was carried out. Specifically, at no time during development was the model adjusted to provide predictions close to *Dutta et al. (2017a)* in reaction to 90% $I_{CaL}$. Therefore, APD shortening with $I_{CaL}$ blockade could be considered a partially calibration criterion, but quantitatively a validation one.

## 14. Validation: propagation at extreme hyperkalemia

For extracellular potassium higher than 9 mM, a fibre was too inexcitable to support the propagation of a full AP. However, a partial activation was observed to propagate throughout the fibre, reaching ca. −30 mV and lasting ca. 50 ms (*Appendix 1—figure 8*). In case of localised hyperkalemia, such as during acute myocardial infarction, it is possible that even such low-peak wave might reactivate nonischemic cells upon exit of the ischemic zone, triggering re-entry and arrhythmia.

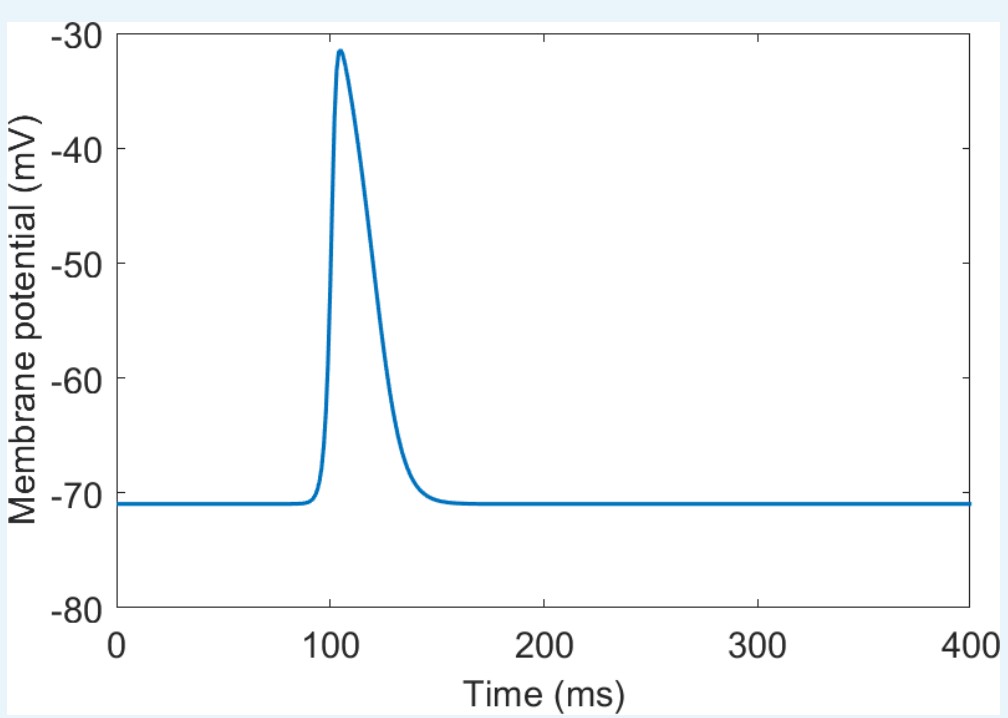

**Appendix 1—figure 8.** Subthreshold fibre activation under extreme hyperkalemia. This trace was measured at the centre of the simulated fibre.

## 15.Simulation methods

### 15.1 Simulation details

Cells were stimulated using a conservative potassium stimulus of −53 mV for 1 ms. Fixed concentrations in the model are: $K_o$ = 5 mM (extracellular potassium), $Na_o$ = 140 mM (extracellular sodium), $Ca_o$ = 1.8 mM (extracellular calcium), $Cl_o$ = 150 mM (extracellular chloride), $Cl_i$ = 24 mM (intracellular chloride).

### 15.1.1 Single-cell simulation protocols

All results are shown at the end of a 1000-beat long pre-pacing train of stimuli. Fixed ionic concentrations were as stated above with two exceptions, where we matched concentrations to the respective studies (this was done both for ToR-ORd and ORd models when generating respective figures). The first exception is the replication of the IV relationship and steady-state inactivation of $I_{CaL}$, based on **Magyar et al. (2000)**, where $Na_o$ = 140.7 mM, $K_o$ = 5.4 mM, $Cl_o$ = 146.2, and $Ca_o$ = 2.5. Particularly, the calcium concentration is of importance, given that it affects the driving force of calcium and thus the IV relationship directly. In addition to the changed values of extracellular concentrations, we fixated the intracellular concentrations of sodium (to 0 mM), potassium (to 153 mM), and chloride (to 130 mM) to match concentrations in the patch-clamp pipette, given that in the long-term, diffusion would distribute the ions in approximately such way. We decided not to fix calcium levels in the model to the pipette concentrations (which was calcium-free), given that calcium entry via $I_{CaL}$ and the resulting release from the SR might last throughout the inactivating pulse, affecting $I_{CaL}$ function in a persistent way.

A second exception is the protocol of induction of early afterdepolarisations (EADs) based on **Guo et al. (2011)**. In this case, $Na_o$ = 137, $Cl_o$ = 148, and $Ca_o$ = 2.

Third, for the P2/P1 protocol (Appendix 1-7), the concentrations were used as in **Fülöp et al. (2004)**: $Cl_i$ = 130 mM, $Cl_o$ = 157 mM, $Na_o$ = 144 mM, $Ca_o$ = 2.5 mM, $K_o$ = 5.6 mM, $K_i$ = 153 mM, $Na_i$ = 0 mM.

### 15.1.2 Hypertrophic cardiomyopathy and hyperkalemia representation

#### 15.1.2.1 Hypertrophic cardiomyopathy

Most of ionic remodelling was based on a comprehensive study by *Coppini et al. (2013)*. All remodelling of HCM cells versus healthy cells was preferentially based on current density, protein expression if current density was not available, or mRNA expression if neither of the previous two measurements was available. The degree of remodelling was based on data means in the Coppini et al. study; where two subunits of a complex underlying a current were altered, an average of the two values corresponding to increase/reduction was used. Based on current density, $I_{NaL}$ was increased by 165%, $I_{CaL}$ by 25%, and $I_{to}$ was reduced by 80%. In addition, consistent with the study, the time constant of fast inactivation (both voltage and calcium-driven) was increased by 35%, and the time constant of slow inactivation was increased by 20%. Based on protein expression, NCX was increased by 50%, $J_{up}$ was reduced by 35%, and $J_{rel}$ by 30%. Based on mRNA, $I_{K1}$ was reduced by 30%, $I_{Kr}$ by 35%, and $I_{Ks}$ by 55%. In addition, $I_{NaK}$ was reduced by 30% (*Passini et al., 2016*), and calcium affinity of troponin was increased slightly by reducing the $K_{trp}$ by 7% (*Robinson et al., 2007*).

#### 15.1.2.2 Hyperkalemia

Extracellular potassium level was raised from 5 mM with an increment of 0.5 mM up to 10 mM. Effective refractory period was determined using S1-S2 protocol, recording the highest S2 coupling interval duration which did not lead to propagation. Hyperkalemic conditions were simulated in a fibre, see below.

### 15.1.3 Fibre simulation

The 1D fibre tissue simulations (hyperkalemia and Appendix 1-11) were conducted using the open-source software CHASTE. The homogeneous 1D tissue fibres had a length of 4 cm consisting of 200 nodes, with a conductivity of 1.64 mS/cm to achieve a diffusion coefficient of 1.171 cm$^2$/s (*Bueno-Orovio et al., 2008*). All the fibres were paced for 20 beats before analysis. For the purpose of measurement of APD or effective refractory period, a stimulus was considered 'propagating' if the peak of membrane potential at the half-point of the fibre was above 0. Stimuli which did not propagate did not have their APD or refractory period measured.

### 15.1.4 Populations of models and drug safety assessment

To assess the performance of the newly developed model for studies of populations of models, we constructed two populations of models using the ToR-ORd model as baseline, and the experimentally-calibrated population of models methodology (*Britton et al., 2013*; *Muszkiewicz et al., 2016*; *Passini et al., 2017*). A total of nine ionic conductances were randomly varied: fast and late $Na^+$ current ($G_{Na}$ and $G_{NaL}$ respectively), transient outward $K^+$ current ($G_{to}$), rapid and slow delayed rectifier $K^+$ current ($G_{Kr}$ and $G_{Ks}$), inward rectified $K^+$ current ($G_{K1}$), $Na^+$-$Ca2^+$ exchanger ($G_{NCX}$), $Na^+$-$K^+$ pump ($G_{NaK}$) and the L-type $Ca^{2+}$ current ($P_{Ca}$). Two variability ranges were considered: [50-150]% and [0–200]% with respect to the baseline values, consistent with the reference study (*Passini et al., 2017*). The number of models passing calibration to experimental biomarkers was 2666 out of 3000 models generated for the [50-150]% population, and 817 out of 3000 for the [0–200]% population.

To assess the suitability of the newly developed model for prediction of drug-induced pro-arrhythmic risk, we also replicated the results of our previous study on human in silico drug trials in population of human models (*Passini et al., 2017*), by testing the same 62 reference compounds at multiple concentrations on the [0–200%] population of models described above. Results were compared in terms of occurrence of drug-induced

repolarisation abnormalities (RA) in the population and TdP score. CredibleMeds (*Woosley and Romero, 2015*) was used as gold standard for TdP risk, dividing the reference compounds in four categories: 1, high risk; 2, possible risk; 3, conditional risk; 4, safe (when not included in CredibleMeds).

All the population of models described above were constructed using the Virtual Assay software (v3.2.1119, 2019 Oxford University Innovation) (*Passini et al., 2017*). Further analysis of the results was performed in Matlab (Mathworks Inc Natwick, MA).

### 15.1.5 Transmurality, 3D simulation, and pseudo-ECG measurement

The formulation of transmurality (representation of endocardial, midmyocardial, and epicardial cells) was retained as in *O'Hara et al. (2011)*, with two exceptions. First, the increase of $I_{CaL}$ conductance from endocardium to midmyocardium was reduced to 2-fold, which corresponds to the mean of the data (*O'Hara et al., 2011*, Figure 10). Second, the increase of $I_{to}$ from endocardium to both midmyocardium and epicardium was reduced to 2-fold to avoid excessive notch following the peak of the AP. Membrane potential traces of the three cell types at 1 Hz pacing are given in *Appendix 1—figure 9*.

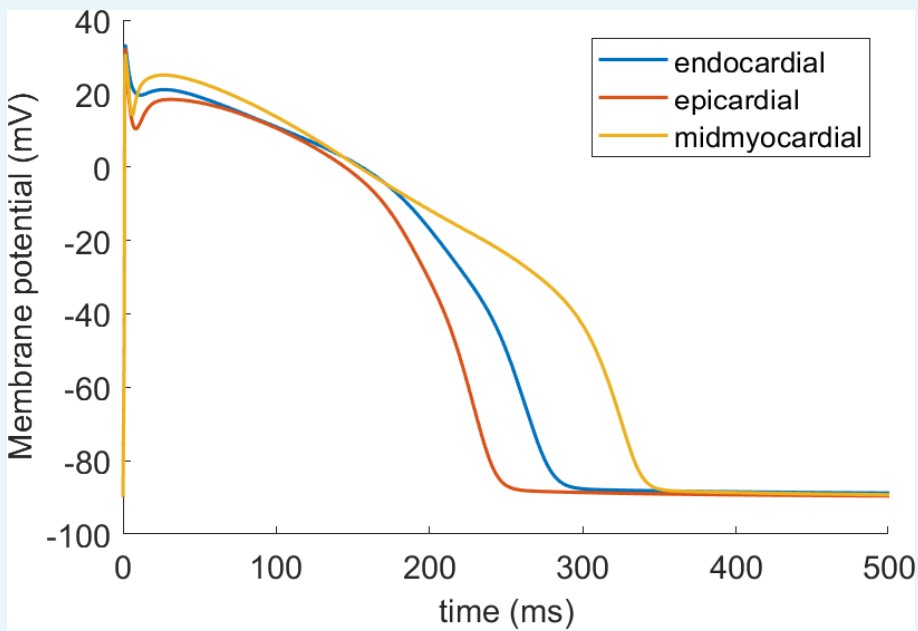

**Appendix 1—figure 9.** Action potential morphology for three cell subtypes at 1 Hz pacing.

The propagation of the electrical activity in the human ventricles was modelled using the heart bidomain equations and solved with the Chaste software (*Pitt-Francis et al., 2009*). A cardiac magnetic resonance (CMR)-informed torso ventricular model was used. The ventricular element size was set to 0.4 mm to ensure numerical convergence of the finite element software Chaste for electrophysiological simulations (*Dutta et al., 2016*; *Pitt-Francis et al., 2009*).

The presented ToR-ORd model was used to represent the membrane kinetics. Transmural and apex to base cell electrophysiological heterogeneities based on experimental and clinical data from *Okada et al. (2011)*; *Drouin et al. (1995)*; *Taggart (2001)*; *Boukens et al. (2015)* were incorporated in the biventricular heart model. Transmural heterogeneities were modelled using three layers as in *Lyon et al. (2018)* of endocardial (45% of the transmural width), mid-myocardial (25%) and epicardial cells (30%) with different AP properties as shown above. Apex-to-base heterogeneities were modelled by including a gradual increase of $I_{Ks}$ conductance from base to apex resulting in APD differences of 25 ms.

Heart fibre directions were generated using the Streeter rule-based method (*Streeter et al., 1969*), and tissue conductivities were set to generate realistic conduction velocities for the ToR-ORd cell model, numerical scheme, and mesh resolution. The longitudinal intracellular conductivity of 1.64 mS/cm in a 1D fibre model resulted in a physiological conduction velocity of ~65 cm/s (*Taggart et al., 2000*). As *Cardone-Noott et al., 2016* used 1.5 mS/cm, intracellular orthotropic conductivities and the axisymmetric extracellular conductivities were obtained by using a scaling factor of 1.0934 to the ones in *Cardone-Noott et al., 2016*.

Sinus rhythm was simulated at 1 Hz using a phenomenological activation model with early endocardial activation sites and a fast endocardial layer representing a tightly-packed endocardial Purkinje network (*Cardone-Noott et al., 2016*) and adapted to the ventricular geometry (*Mincholé et al., 2019*).

Simulation of the pseudo-ECG was computed by calculating the extracellular potentials in the virtual standard 12-lead electrode positions (*Lyon et al., 2018*) from the ventricular geometry following the dipole model (*Gima and Rudy, 2002*) as:

$$\phi(\boldsymbol{e}) = \int_\Omega \left( D\nabla V_m \cdot \left( \nabla \frac{1}{\|\boldsymbol{r}-\boldsymbol{e}\|} \right) \right) d\boldsymbol{r}$$

where $\boldsymbol{e} = (e_x, e_y, e_z)$ are the electrode position coordinates, $D$ is the diffusion tensor, and $V_m$ is the membrane potential. The integral is calculated over the whole myocardium volume, $\Omega$.

### 15.1.6 Timely killing of crashing simulations during multiobjective GA

We found it crucial to modify the behaviour of @ode15s in our simulations, using Matlab ODE Events to limit 1 s of simulated time to 5 s of runtime (with the runtime being <0.5 s in normal conditions), killing a simulation upon exceeding the limit. When parameters of the models are randomly perturbed during GA fitting, it is possible to achieve a combination which leads to model instability and eventual crash. However, as ode15s attempts to reduce time step in such a case, it takes up to 6 hr to crash, blocking CPU cores and stalling the whole generation. With the timely killing of unstable simulations, we could run 30 generations with population size of 2500 in ca. 30 hr on an Azure virtual machine with 64 cores, using parallel fitness evaluation.

## 15.2 Evaluation pipeline and HTML reporter

In addition to the model code itself, we also provide a model evaluation pipeline. This allows simulation of the single-cell calibration and validation criteria using a standard model function (I.e. all except $I_{CaL}$ properties such as I-V relationship, or disease models, where modified model codes are used.). After the simulation of a comprehensive set of protocols, an HTML report is produced (*Appendix 1—figure 10*), providing a clickable mapping between criteria and figures showing how the model fulfils them. Icons adjacent to the criteria allow rapid assessment of whether the model behaves well in the given criterion. Any model with appropriate interface of inputs and outputs can be simulated. It is particularly easy to simulate variants of the ToR-ORd model, for example with changed conductances of currents, as the same structure of parameters that is passed to the model simulation wrapper can be passed to this evaluation pipeline. Given that the evaluation code and subsequent report generation are fully automated, it is easy to extend the pipeline to compute and visualise other evaluation criteria.

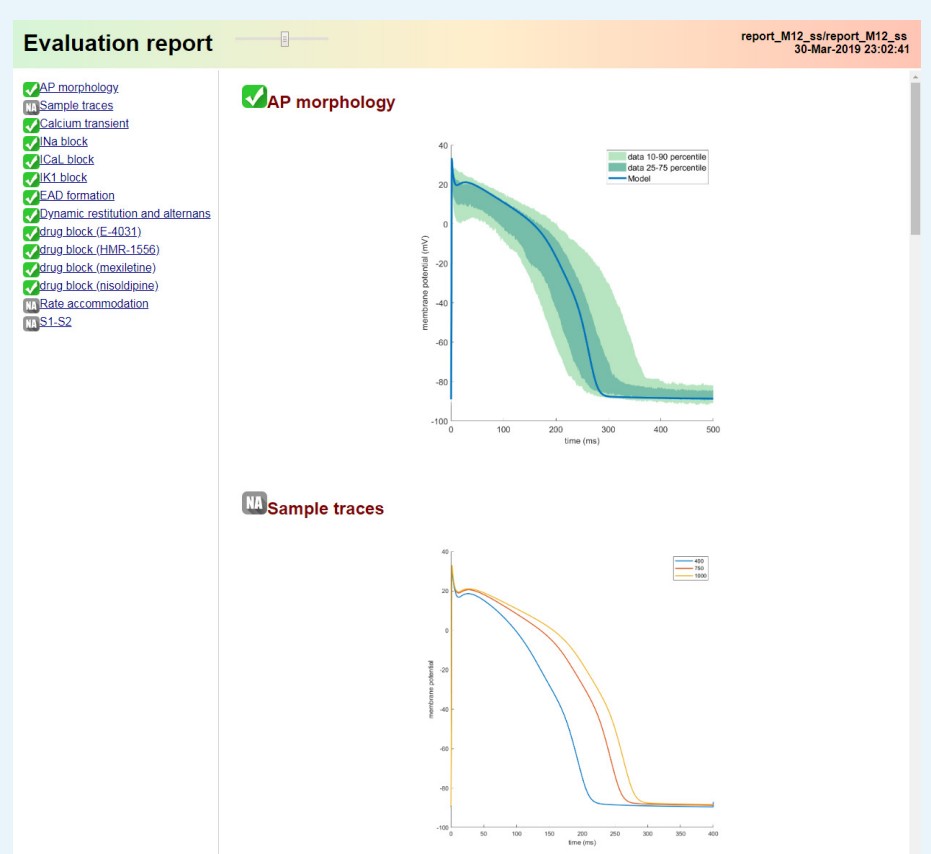

**Appendix 1—figure 10.** Evaluation pipeline. The left menu shows a list of evaluation criteria with the adjacent icon indicating whether they are fulfilled by the model. The right pane contains figures visualising the model performance in the criteria. The figures can be either viewed by scrolling or by clicking the corresponding entry in the left menu. The report header contains a timestamp of the report generation and a slider which controls the size of figures in the right pane. The whole report is written in HTML, as automatically generated from Matlab code, and can be viewed in any internet browser.

The whole process of report generation (and storage of included plots) can be run separately for calibration and validation criteria (with separately stored figures), facilitating unbiased model development, where the model can be assessed using the calibration criteria only, without the user observing any validation results. It is nevertheless possible to also generate a report for only validation criteria, or both calibration and validation.

Below are given the criteria evaluated, along with how the rating of the model is defined (PASS = good behaviour in a given criterion, FAIL = problematic behaviour, NA = feature not automatically compared to data, usually because of difficulty of accurate rating; this code is also used when a plot is only an illustration, such as examples of action potential shape at different pacing frequencies). Unless specified otherwise, FAIL is given when PASS is not fulfilled.

## 15.2.1 Calibration criteria

1. Action potential morphology. PASS ~ AP from 10 to 500 ms is within the 10–90% quantile range of the Szeged dataset. The first 10 ms are ignored so as not to limit the model in its peak potential, as this is strongly modulated by cell coupling (the available AP data are based on small-tissue samples).
2. Sample traces showing the AP and calcium transient at different pacing frequencies. Always NA, this is just an illustrator figures useful for an eyeballing-type of assessment.

3. Calcium transient properties (time to peak, duration at 90% recovery. PASS ~ both features are within estimated standard deviation of data given in *Coppini et al. (2013)*.
4. Negative inotropy of 50% $I_{Na}$ + 50% $I_{NaL}$ block. PASS if such treatment induces a reduction in transient amplitude.
5. APD shortening with 50% $I_{CaL}$ block. PASS if such treatment induces APD shortening.
6. The effect of 50% $I_{K1}$ block on membrane potential. PASS if such treatment induces depolarisation.
7. EAD formation. PASS if dV/dt exceeds 0.05 after 50 ms of the AP. This is assessed at 4000 ms bcl, 85% $I_{Kr}$ block, and concentrations as given in section 1.3.1.
8. Alternans formation. PASS if alternans of amplitude more than 5 ms is formed at any frequency faster than 300 ms base cycle length. The plots generated can be also used to assess dynamic restitution of APD.

Conduction velocity is not assessed, as it requires simulating a fibre using other tools than Matlab.

### 15.2.2 Validation criteria

1. Four simulated drug blocks from *Dutta et al. (2017a)*:

   ○ E-4031 ($I_{Kr}$ blocker)
   ○ HMR-1556 ($I_{Ks}$ blocker)
   ○ Mexiletine ($I_{NaL}$, $I_{Kr}$, $I_{CaL}$ blocker)
   ○ Nisoldipine ($I_{CaL}$ blocker)
2. Each is simulated separately, and each is scored as PASS if the model prediction is within the standard deviation of the data.
3. APD accommodation. Always NA, as it is challenging how to define agreement with data exactly.
4. S1-S2 protocol. Always NA, just a comparison to data is given.

Effects of hyperkalemia, POM and drug safety testing, or 3D heart simulation are not assessed automatically, as they requires simulation tools other than Matlab.

## 15.3 Summary of updated equations

Below are the equations which have been added and/or changed in the ORd model. The equations follow the naming convention of the ORd model with the exception of $I_{Na}$, $I_{(Ca)Cl}$, $I_{Clb}$, and $I_{K1}$, which are based on different models. Membrane potential is expressed as $V$.

### 15.3.1 $I_{Na}$

The *Grandi et al. (2010)* model of $I_{Na}$ was used as a baseline, which was extended to account for the effect of CaMKII via modulation of the $h$ gate (a shift by 6 mV) and the 1.46 times increase of the time constant of $j$ (recovery from inactivation), consistent with (*O'Hara et al., 2011*). The current was included within the membrane adjacent to the main cytosolic pool as in ORd. The equations underlying describing the Grandi model were extracted from the model source code.

$$m_\infty = \frac{1}{\left(1 + e^{-\frac{56.86+V}{9.03}}\right)^2}$$

$$\tau_m = 0.1292 \cdot e^{-\left(\frac{V+45.79}{15.54}\right)^2} + 0.06487 \cdot e^{-\left(\frac{V-4.823}{51.12}\right)^2}$$

$$\frac{dm}{dt} = \frac{m_\infty - m}{\tau_m}$$

$$a_h = \begin{cases} 0.057 \cdot e^{-\frac{V+80}{6.8}} & \text{for } V < -40 \text{ mV} \\ 0 & \text{otherwise} \end{cases}$$

$$b_h = \begin{cases} 2.7 \cdot e^{0.079 \cdot V} + 3.1 \cdot 10^5 \cdot e^{0.3485 \cdot V} & \text{for } V < -40 \text{ mV} \\ \frac{0.77}{0.13 \cdot \left(1 + e^{-\frac{V+10.66}{11.1}}\right)} & \text{otherwise} \end{cases}$$

$$\tau_h = \frac{1}{ah + bh}$$

$$h_\infty = \frac{1}{\left(1 + e^{\frac{V+71.55}{7.43}}\right)^2}$$

$$\frac{dh}{dt} = \frac{h_\infty - h}{\tau_h}$$

$$a_j = \begin{cases} \frac{\left(-2.5428 \cdot 10^4 \cdot e^{0.2444 \cdot V} - 6.948 \cdot 10^{-6} \cdot e^{-0.04391 \cdot V}\right) \cdot (V+37.78)}{1 + e^{0.311 \cdot (V+79.23)}} & \text{for } V < -40 \text{ mV} \\ 0 & \text{otherwise} \end{cases}$$

$$b_j = \begin{cases} \frac{0.02424 \cdot e^{-0.01052 \cdot V}}{1 + e^{-0.1378 \cdot (V+40.14)}} & \text{for } V < -40 \text{ mV} \\ \frac{0.6 \cdot e^{0.057 \cdot V}}{1 + e^{-0.1 \cdot (V+32)}} & \text{otherwise} \end{cases}$$

$$\tau_j = \frac{1}{aj + bj}$$

$$j_\infty = \frac{1}{\left(1 + e^{\frac{V+71.55}{7.43}}\right)^2}$$

$$\frac{dj}{dt} = \frac{j_\infty - j}{\tau_j}$$

$$hp_\infty = \frac{1}{\left(1 + e^{\frac{V+71.55+6}{7.43}}\right)^2}$$

$$\frac{dhp}{dt} = \frac{hp_\infty - hp}{\tau_h}$$

$$\tau_{jp} = 1.46 \cdot \tau_j$$

$$\frac{djp}{dt} = \frac{j_\infty - jp}{\tau_{jp}}$$

$$G_{Na} = 11.7802$$

$$I_{Na} = G_{Na} \cdot m^3 \cdot \left((1 - f_{INap}) \cdot h \cdot j + f_{INap} \cdot hp \cdot jp\right)$$

where $hp$ is the phosphorylated $h$ gate, $jp$ is the phosphorylated $j$ gate, and $f_{INap}$ is the fraction of CaMKII-phosphorylated sodium channels.

### 15.3.2 $I_{NaL}$

As the ORd model assumes the time constant of $I_{NaL}$ activation to be the same as the one of $I_{Na}$, $\tau_m$ of $I_{NaL}$ in ToR-ORd was likewise set to $\tau_m$ of the extended Grandi $I_{Na}$ model as described above.

$$\tau_{m,L} = 0.1292 \cdot e^{-\left(\frac{V+45.79}{15.54}\right)^2} + 0.06487 \cdot e^{-\left(\frac{V-4.823}{51.12}\right)^2}$$

In addition, the conductance was changed:

$$G_{NaL} = 0.0279$$

### 15.3.3 I$_{CaL}$

#### 15.3.3.1 Activation

Steady-state activation was reformulated as the following Gompertz function restricted to [0,1].

$$d_\infty = \left(1.0763 \cdot e^{-1.007 \cdot e^{-0.0829 \cdot V}} \text{ for } V \leq 31.4978 \,|\, 1 \text{ otherwise}\right)$$

#### 15.3.3.2 Localisation

20% of I$_{CaL}$ was placed within the membrane adjacent to the main cytosolic ionic pool ($I_{CaL,i}$); 80% is included within membrane adjacent to the 'subspace' compartment ($I_{CaL,SS}$). Total I$_{CaL}$ (the sodium and potassium currents carried by the channel are treated analogously) is given as follows:

$$I_{CaL} = 0.8 \cdot I_{CaL,SS} + 0.2 \cdot I_{CaL,i}$$

The two parts of I$_{CaL}$ differ in the driving force (which differs between subspace and main ionic pool based on potentially different ionic concentrations), and $n$, the variable determining the degree of calcium inactivation. We separated the original $n$ variable into $n_{SS}$ (fraction of calcium-inactivated I$_{CaL}$ channels in the junctional subspace) and $n_i$ (fraction of calcium-inactivated I$_{CaL}$ channels in the main cytosolic ionic pool). These are updated as $n$ in the original model, except that $n_i$ is updated using the calcium concentration in the main cytosolic pool, rather than in junctional subspace.

#### 15.3.3.3 Driving force

Driving force is modelled using the Goldman-Hodgkin-Katz equation as in the original model, but this is computed separately for I$_{CaL}$ adjacent to junctional subspace compartment and the main cytosolic ionic pool, using corresponding ionic concentrations and activity coefficients. Activity coefficient $\gamma_{X,SS}$ (activity of ionic specie X, such as Ca, Na, or K in the junctional subspace compartment) is computed dynamically based on Davies equation:

$$\gamma_{X,SS} = e^{-A \cdot z_X^2 \cdot \left(\frac{\sqrt{I}}{1+\sqrt{I}} - 0.3 \cdot I\right)},$$

where $A = 1.82 \cdot 10^6 \cdot (74 \cdot 310)^{-1.5}$, $z_X$ is the charge of X, and $I$ is the ionic strength:

$$I = 0.5 \cdot \sum_i c_i \cdot z_i^2,$$

where $c_i$ is the molarity (I.e. the concentrations used throughout the model, which are given in mM, are divided by 1000 to convert them to M.) of the i-th ion (ToR-ORd takes into account calcium, sodium, potassium, and chloride) in the given compartment, and $z_i$ is its charge.

In total, three values of ionic strength are used in ToR-ORd for one ionic specie, one for each of the following compartments: extracellular, main intracellular pool, and junctional subspace. This then yields three corresponding activity coefficients for the given ionic specie.

#### 15.3.3.4 Other changes

$$jca_\infty = \frac{1}{1 + e^{-\frac{V+18.08}{2.7916}}}$$

$$\frac{djca}{dt} = \frac{jca_\infty - jca}{\tau_{jca}}$$

$$k_{=2,n} = 500 \text{ (used both in computation of } n_i \text{ and } n_{ss})$$

$$P_{Ca} = 8.3757 \cdot 10^{-5}$$

### 15.3.4 $I_{To}$

$$G_{to} = 0.16$$

### 15.3.5 $I_{Kr}$

The Lu-Vandenberg (**Lu et al., 2001**) model was used, with structure given in **Figure 1B** of the main manuscript. The model is used as published, except with slightly increased $\alpha_1, \beta_i$, and different conductance. In this section, $I$ without a subscript stands for the fraction of channels in the inactivated state, not for a current.

$$F = 96485$$

$$R = 8314$$

$$T = 310$$

$$vfrt = V \cdot \frac{F}{R \cdot T}$$

$$\alpha = 0.1161 \cdot e^{0.299 \cdot vfrt}$$

$$\beta = 0.2442 \cdot e^{-1.604 \cdot vfrt}$$

$$\alpha_1 = 1.25 \cdot 0.1235$$

$$\beta_1 = 0.1911$$

$$\alpha_2 = 0.0578 \cdot e^{0.971 \cdot vfrt}$$

$$\beta_2 = 0.349 \cdot 10^{-3} \cdot e^{-1.062 \cdot vfrt}$$

$$\alpha_i = 0.2533 \cdot e^{0.5953 \cdot vfrt}$$

$$\beta_i = 1.25 \cdot 0.0522 \cdot e^{-0.8209 \cdot vfrt}$$

$$\alpha_{C2toI} = 0.52 \cdot 10^{-4} \cdot e^{1.525 \cdot vfrt}$$

$$\beta_{ItoC2} = \frac{\beta_2 \cdot \beta_i \cdot \alpha_{C2ToI}}{\alpha_2 \cdot \alpha_i}$$

$$\frac{dC_0}{dt} = C_1 \cdot \beta - C_0 \cdot \alpha$$

$$\frac{dC_1}{dt} = C_0 \cdot \alpha + C_2 \cdot \beta_1 - C_1 \cdot (\beta + \alpha_1)$$

$$\frac{dC_2}{dt} = C_1 \cdot \alpha_1 + O \cdot \beta_2 + I \cdot \beta_{ItoC2} - C_2 \cdot (\beta_1 + \alpha_2 + \alpha_{C2toI})$$

$$G_{Kr} = 0.0321$$

$$I_{Kr} = G_{Kr} \cdot \sqrt{\frac{K_o}{5}} \cdot O \cdot (V - E_K)$$

$K_o$ stands for extracellular potassium concentration.

### 15.3.6 $I_{Ks}$

$$G_{Ks} = 0.0011$$

### 15.3.7 $I_{K1}$

The formulation below is based on **Carro et al. (2011)**, updated to take into account that extracellular potassium in our model is 5 mM, rather than 5.4.

$$a_{K1} = \frac{4.094}{1 + e^{0.1217 \cdot (V - E_K - 49.934)}}$$

$$b_{K1} = \frac{15.72 \cdot e^{0.0674 \cdot (V - E_K - 3.257)} + e^{0.0618 \cdot (V - E_K - 594.31)}}{1 + e^{-0.1629 \cdot (V - E_K + 14.207)}}$$

$$K1_{SS} = \frac{a_{K1}}{a_{K1} + b_{K1}}$$

$$G_{K1} = 0.6992$$

$$I_{K1} = G_{K1} \cdot \sqrt{\frac{K_o}{5}} \cdot K1_{SS} \cdot (V - E_K)$$

### 15.3.8 $I_{NaCa}$

$$G_{NCX} = 0.0034$$

Fraction of NCX in the junctional subspace was set to 0.35.

### 15.3.9 $I_{NaK}$

$$P_{NaK} = 15.4509$$

### 15.3.10 $I_{(Ca)Cl}$

This current is placed within the junctional subspace

$$I_{(Ca)Cl} = \frac{0.2843 \cdot (V - E_{Cl})}{1 + \frac{0.1}{Ca_{ss}}}$$

### 15.3.11 I_Clb

$$I_{Clb} = 1.98 \cdot 10^{-3} \cdot (V - E_{Cl})$$

### 15.3.12 I_Kb

$$x_{Kb} = \frac{1}{1 + e^{-\frac{V - 10.8968}{23.9871}}}$$

$$I_{Kb} = 0.0189 \cdot x_{Kb} \cdot (V - E_K)$$

### 15.3.13 I_Cab

$$vffrt = V \cdot F \cdot \frac{F}{R \cdot T}$$

$$vfrt = V \cdot \frac{F}{R \cdot T}$$

$$I_{Cab} = 5.9194 \cdot 10^{-8} \cdot 4 \cdot vffrt \cdot \frac{\gamma_{Cai} \cdot Ca_i \cdot e^{2 \cdot vfrt} - \gamma_{Cao} \cdot Ca_o}{e^{2 \cdot vfrt} - 1}$$

The second half of the right hand side (from *vffrt* on, inclusive) is the Goldman-Hodgkin-Katz driving force; $\gamma_{Cai}$ is the activity coefficient of intracellular calcium (in the main ionic pool), and $\gamma_{Cao}$ is the activity coefficient of extracellular calcium.

### 15.3.14 I_Nab

$$P_{Nab} = 1.9238 \cdot 10^{-8}$$

### 15.3.15 J_rel

$$K_{\infty, rel} = 1.7$$

$$J_{rel,NP,\infty} = \frac{\alpha_{rel} \cdot (-I_{CaL})}{1 + \left(\frac{K_{\infty,rel}}{Ca_{JSR}}\right)^8}$$

$$J_{rel,P,\infty} = \frac{\alpha_{rel,CaMK} \cdot (-I_{CaL})}{1 + \left(\frac{K_{\infty,rel}}{Ca_{JSR}}\right)^8}$$

$$J_{rel} = 1.5378 \cdot \left(1 - \phi_{rel,CaMK}\right) \cdot J_{rel,NP} + \phi_{rel,CaMK} \cdot J_{rel,CaMK}$$

### 15.3.16 J$_{up}$

$$J_{up,NP} = \frac{0.005425 \cdot Ca_i}{0.00092 + Ca_i}$$

$$J_{up,P} = \frac{2.75 \cdot 0.005425 \cdot Ca_i}{0.00092 - 0.00017 + Ca_i}$$

$$J_{leak} = \frac{0.0048825 \cdot Ca_{NSR}}{15}$$

