## [Decision Letter]

**Acceptance summary:**

This work constitutes a significant advancement in the field of theoretical cardiac electrophysiology. Not only is the proposed mathematical model of the human ventricular myocyte an improvement over existing models, in terms of its correspondence with experimental data and its potential to accelerate therapeutic developments in human cardiac electrophysiology; in addition, the calibration and validation of this model also exemplifies a rigorous methodology that will hopefully become standard in this field.

**Decision letter after peer review:**

Thank you for submitting your article "Development, calibration, and validation of a novel human ventricular myocyte model in health, disease, and drug block" for consideration by *eLife*. Your article has been reviewed by three peer reviewers, and the evaluation has been overseen by José D. Faraldo-Gómez as Reviewing Editor and Naama Barkai as the Senior Editor. The following individuals involved in the review of your submission have agreed to reveal their identity: Thomas Hund (Reviewer #1); Molly Maleckar (Reviewer #3).

Although it is customary for *eLife* to condense reviewers' reports into a concise decision letter, in this case the Reviewing Editor believes it would be best to enclose these reports as originally submitted. Based on these reports, we would like to invite you to submit a revised version of your manuscript that addresses the questions and concerns raised.

Reviewer #1:

This paper presents a modified model of the human ventricular action potential based on a published model from the Rudy group (O'hara Rudy – ORd model). Changes were made to improve the ability of the model to reproduce different aspects of the action potential – especially the plateau potential and APD accommodation. To achieve this goal, Tomek et al. modified the L-Type calcium channel, as well as replacing the formulation of the sodium channel and rapid delayed rectifier channel. The L-Type channel was modified by re-deriving the ionic activity, and treating it as variable in subspaces and time. The L-Type was then also refit using the activation curve normalized to the GHK driving force, rather than the Nerst driving force as was done previously. Additionally, the formulation of the sodium channel was replaced with a version from Grandi et al., 2010, and I_Kr_ was replaced with a version from Lu et al., 2001. The changes were than fit to data from the original O'Hara Rudy paper and elsewhere. Finally, the model was validated against APD accommodation from O'Hara, 2011, drug safety predictions, and more, including ECG results from whole heart simulations. This model introduces several notable improvements to the O'Hara Rudy model that will be of interest to mathematical modeling researchers.

1) Chloride current is added to the model without fitting or showing how it affects the model. Additionally, it would be helpful to have a justification of why these currents were added, as well as validated.

2) In Figure 2, inclusion of the other main dataset used by O'Hara to fit the L-Type channel would help in the comparison of the new L-Type formulation to the old formulation. (Fulop et al., 2004).

3) In Figure 5, the fit of the fifth drug, BaCl_2_, would still be interesting to see as it was used in both O'Hara and Dutta et al., 2017, despite BaCl_2_ having off-target effects.

4) Please review all equations for typos.

Reviewer #2:

Tomek et al. have put a tremendous amount of effort into improving the O'Hara et al. mathematical model of the human ventricular myocyte. The correspondence of the new model with experimental data is quite impressive, and the authors have done a laudable job of separating the calibration and validation steps in model development. Because these steps have often been combined and/or blurred in previous work in this field, the study provides additional rigor by separating the two steps. Moreover, given the central importance of the O'Hara et al. model in the field, there is a need for improvements, and this work most likely represents a major step in the right direction. These are all reasons to like the paper.

All of that being said, there are certain aspects of the manuscript that are rather confusing and need to be improved. Although the comments below may seem excessive to the authors, it did in fact take multiple readings and considerable additional thinking for this reviewer to fully comprehend some of what the authors had done. It can be a difficult goal to achieve, but the authors should aim for a paper that can be readily understood, even by non-experts, after a single reading. The manuscript doesn't meet that standard at present.

1) One of the most interesting aspects of the manuscript is the discussion of activity coefficients, I_CaL_ current-voltage plots, and the extraction of activation curves (subsection “In-depth revision of the L-type calcium current”). The reviewer absolutely concurs that because many formulations have been inherited from previous models, often without rigorous examination of these formulations, it is entirely appropriate to examine these assumptions and re-formulate currents if necessary. However, this discussion is ultimately unconvincing, and the presentation needs to be modified for this section to be fully understood.

1a) The discussion of the activity coefficients is interesting and will be informative to readers unfamiliar with this primary literature. However, although the authors make a compelling case that different activity coefficients, i.e. closer to 0.6, should be used, the manuscript leaves out the most important question, namely: how do these changes affect the I-V curves?

1b) Speaking of I-V curves, examination of Figure 1D makes a convincing case that the new model has a different shape and matches the experimental data better. However, it's not clear why this occurs – what allows this better fit? Is this due to the changes in activity coefficients? Is it just from the better activation curve shown in Figure 1C? Another fact that can affect this curve is the relative permeability of the channel to Na and K compared with Ca. It's odd that this is never addressed in the manuscript. Did the authors modify this to try to fit the curves better, or were these numbers considered fixed based on original data?

1c) The term "driving force" is used in the manuscript in a way that is inconsistent with prior literature. One of the appeals of Ohmic formulations is that driving force is expressed in units of volts, and conductance can be derived from current plots by simple division. But with a GHK formulation, things are not so simple. The term called "driving force" here does not appear to be in units of volts, so what exactly is this term representing?

1d) Finally, on this same topic, the Discussion states: "Activation curve of the current in previous cardiac models was based on the use of Nernst driving force in experimental studies, but the models then used Goldman-Hodgkin-Katz driving force to compute the current." Again, it's not completely clear how the authors derive one set of numbers (i.e. activation at defined voltages) from a different set of numbers (i.e. current at those same voltages). As I have been writing this, it has occurred to me that perhaps the units of driving force are less relevant, as long as one set of numbers can be multiplied by a different set of numbers to ultimately produce current. However, the fact that I've had to think so hard about this demonstrates that the manuscript needs to do better. Because the I_CaL_ formulation is claimed in the Discussion as "the greatest theoretical contribution of this work," the explanations here need to be crystal clear. Perhaps the supplement should contain flow charts of the conventional method for extracting I_CaL_ activation versus how the new approach improves this, as well as plots showing how the changes to the activity coefficients influence the IV curves.

2) The discussion of Na current blockade and inotropy is relatively weak. Perhaps the authors need to reproduce some original experimental data, or, at the very least, provide some numbers extracted from the earlier studies. The description of Figure 3 discusses the "negative inotropy" previously reported with Na channel blockers, but with no numbers provided. The change in Ca transient amplitude shown in Figure 3C is very small, probably undetectable experimentally. If experiments have reported a 50% decrease in contraction strength, I would say that neither model is consistent, although ToR-ORd is slightly better. Obviously the heatmaps in 3E and 3F provide additional information, but in general here the correspondence with data can be discussed in a more quantitative way. In the Discussion, the phrase "good response of the ToR-ORd model to sodium blockade" sticks out as rather vague.

2a) Again on the Na channel block-inotropy issue. The authors have an opportunity to obtain new mechanistic insight here given that the response of the two models is different. My suspicion is that it might have less to do with the AP morphology, as speculated in the Discussion, and more to do with intracellular [Na] regulation. However, since mechanisms are not explored, we don't know which idea is correct. It might be considered beyond the scope of this study to obtain this mechanistic insight, but it's an interesting question, and the new model provides a means for addressing it.

3) The description of hyperkalemia simulation in Figure 8 should be improved. First, the text claims that Figure 8A shows a "progressive" increase in resting potential and slowing of upstroke velocity. But since the figure only shows 2 examples, it's hard to see that this is progressive. Second, PRR is used in this description but has not been previously defined. It took a few seconds to figure out what the authors meant. Is it really necessary to have an abbreviation for a term that's only used once in the manuscript?

4) The last paragraph of the Discussion, discussing future directions, should be modified. This paragraph states: "Similarly to most existing cardiac models, the equations governing the release depend directly on the L-type calcium current, rather than on the calcium concentration adjacent to the ryanodine receptors, which is the case in cardiomyocytes. Future development of the ryanodine receptor model and calcium handling will extend the applicability of the model.…"

The first sentence is correct, the second is not. The challenge is not to improve the representation of the ryanodine receptor. This will not fix the problems; the problems arise from the fact that Ca release in cells is controlled locally rather than globally – this is why whole cell models, that use a single variable to describe ryanodine receptor gating, need to use shortcuts such as making release directly dependent on I_CaL_. This general problem was rigorously analyzed by Stern way back in 1992 (PMID: 1330031). For more discussion, see also PMIDs: 15465866, 20346962, 21586292.

It's great to discuss potential future improvements to the EC coupling part of the model, but these discussions should acknowledge that the issue is not just improving the RyR model, but that stochastic triggering of locally-controlled Ca sparks will need to be described.

Reviewer #3:

Mature models and simulation techniques in human cardiac electrophysiology can now be exploited to accelerate therapeutic development and validation e.g. devices, drug development/cardiotoxicity. The authors identify key weaknesses in a model of the human ventricular action potential (ORd, which has been chosen previously by an expert-led initiative to represent the human ventricular AP in validation studies incorporating models and simulation), and present an updated (novel) model of the human AP, the ToR-ORd, to address these.

Overall, the work addresses an important need: in order to effectively translate value from models and simulations in the hoped-for context of human ventricular electrophysiology, the primary model's weaknesses must absolutely be addressed for utility, and so I would consider this work far beyond incremental and a very useful contribution in terms of model to the field at this juncture. The work also nicely leverages a plethora of prior experimental work using direct electrophysiological measurements for calibration and validation. In general, the work is very well organized and well-written.

Comments:

- Cogent, well-motivated Introduction. Weaknesses of ORd (AP plateau potential, APD adaptation and response to sodium current block) and solutions (L-type calcium, excitation contraction coupling and hERG current re-assessment and reformulations) clear.

- Overall goal to design, develop, calibrate and validate the novel ToR-ORd model, with aim of reproducing all key depolarization, repolarization and calcium dynamics properties in healthy human ventricular cells and when these are under drug block, as well as in diseased conditions e.g. hyperkalemia and hypertrophic cardiomyopathy is both timely and useful.

- Importantly, calibration and validation processes are independent, using independent datasets

- "The I_CaL_ current was deeply revisited, particularly with respect to its driving force, based on biophysical principles." – well-motivated, needed, seldom done, valuable to other models. "This allows accurate representation of the driving force when ionic concentrations are disturbed,"

- Really encouraging to see disease-based, multi-scale validation of this nature. Impact for translational work made abundantly clear.

Concerns/areas to address:

1) Given the journal's wide appeal, there is a need to better introduce the strategy and process going into the Materials and methods section, from "We initially performed the evaluation of the ORd model (O'Hara et al., 2011) against calibration criteria…". While I appreciated the simplicity of explanation in general, this guiding summary can be improved to clarify what was initially done and why and how the strategy developed

2) It's also not immediately clear what the calibration criteria actually are (Table 1) – I assume that, in addition to the listed references, that these exist in a supplement somewhere and it would be great to point the reader to this

3) "Simulations with the existing versions of the ORd model failed to fulfil key criteria such as AP morphology, calcium transient duration, several properties of the L-type calcium current, negative inotropic effect of sodium blockers, or the depolarising effect of I_K1_ block" – where is this shown? Is there a supplementary figure?

4) In-depth revisions based on fundamental physical principles is a much-needed process. Some motivation and/or discussion of the state of the art in this particular field (i.e. how some attention to the underlying biophysics was laid by the wayside in model redevelopment, experimental assimilation, and subsequent versioning, and conjectures as to why e.g. computational tools like versioning tools did not exist, would be appropriate

5) Given the progressive, savvy lean of this work, it seems reasonable to ask why authors did not consider making the code available in other formats, e.g. python as well, or at least justify/mention the relevance of language and tools for reuse/reproducibility/versioning. Similarly, reference to other recent forays into systematic model improvement, e.g. functional curation and web lab, seems warranted

6) Results: "This [ToR-ORd model yields negative I_CaL_ values in such conditions] is a direct consequence of the updates to the extracellular/intracellular calcium activity coefficients, which supports its credibility and it is important for cases of elevated I_CaL_, such as under ß-adrenergic stimulation." – while this is likely true, it also seems likely a throwaway – were any simulations run using the model to demonstrate the importance in this vein? Important how?

7) Results/Figure 3: Besides noting that the ToR-ORd is consistent with observed negative inotropy of sodium blockers, it is unclear from text how updated model compares in terms of available experimental data, i.e. "A mild increase in inotropy may be achieved only under near-exclusive I_Na_ block." – or whether this comparison is available at all.

8) Figure 5A, B, C – the slope of the APD90 dependence on BCL for the ToR-ORd differs consistently from that of the data for BCLs > 1s – any musings as to mechanism? Either briefly in main text or supplement.

9) Figure 6C: would strengthen the case for the ToR-ORd further to compare restitution for ORd in a fibre under same protocol?

10) Seems like suitability of ToR-ORd for drug safety testing is clear – and discussion around this is adequate and encouraging. However, there still does not seem to be a marked improvement in the example given over the ORd (Figure 7), particularly in terms of false negatives. Authors note "further improvement to the score is likely with development of a population optimised towards drug safety assessment" – so it will be interesting to see results there.

---

## [Author Response]

Reviewer #1:1) Chloride current is added to the model without fitting or showing how it affects the model. Additionally, it would be helpful to have a justification of why these currents were added, as well as validated.

The chloride currents were parametrised exactly as in Grandi et al., 2010, (only the conductances were adjusted). We have now added a more detailed description of their effect and justification in the Materials and methods.

“The motivation to add these currents was to facilitate the shaping of post-peak AP morphology (via I_(Ca)Cl_), with I_Clb_ playing a dual role stemming from its reversal potential of ca. -50 mV. It slightly reduces plateau potentials during the action potential, but during the diastole, it depolarises the cell slightly, improving the reaction to I_K1_ block as explained in the next subsection.”

2) In Figure 2, inclusion of the other main dataset used by O'Hara to fit the L-Type channel would help in the comparison of the new L-Type formulation to the old formulation. (Fulop et al., 2004).

Thank you for the comment. We have now simulated this protocol and added it as Appendix 1—figure 4, with a mention in the main manuscript. Simulated data with the ToR-ORd are similar to the experimental recordings and the ones obtained with the ORd in that a) the short pulse-curve lies above the long-pulse curve, b) the shorter coupling interval is associated with less current availability. Both models yield an offset with respect to the experimental data. This could be due to a) Difference in I_CaL_ behaviour/density and/or calcium loading (affecting inactivation) between the Fülöp (P2/P1 protocol) and Magyar (the main basis for the I_CaL_ model) studies i.e., it is possible that even the living cells could manifest different properties. It could be also linked to mechanisms of I_CaL_ inactivation not represented in the model (e.g. the inactivation by the calcium flow through the channel, represented e.g., by Mahajan et al., 2008). If we accelerate I_CaL_ inactivation, we observed that the model stopped being capable of manifesting EADs, which is a very important feature, and losing it would make the model less useful than in the current form. We speculate that a more complex model structure as well as calcium-sensitive SR release may be required to achieve a model with sufficiently fast inactivation, but which is nevertheless capable of manifesting EADs. This is future work that we will most likely work on in the future, but it is currently well beyond the scope of this paper.

3) In Figure 5, the fit of the fifth drug, BaCl_2_, would still be interesting to see as it was used in both O'Hara and Dutta et al., 2017, despite BaCl_2_ having off-target effects.

Thank you for this comment. We have now included an explanation in the manuscript as to why we chose not to include a comparison to BaCl_2_ (Main text subsection “Validation: Drug-induced effects on rate dependence of APD” and Appendix 1 subsection “Validation: Drug block and APD”). This is fundamentally due to the uncertainty concerning its multichannel effects on several potassium currents, in addition to I_K1_. It was previously shown that BaCl_2_ blocks also I_Kr_ and I_Ks_ at the used concentration (Weerapura et al., 2000; Gibor et al., 2004). Furthermore, as shown in (DOI 10.1371/journal.pone.0053255), even a lower dose of BaCl_2_ prolongs the AP plateau and early phase of repolarization, thus at membrane potentials at which I_K1_ is not activated. This supports the supposition that BaCl_2_ is likely to block I_Kr_ and I_Ks_, in addition to I_K1_, but to an uncertain degree. Furthermore, pure block of I_K1_ in the ORd model results in APD prolongation from around -50mV only, which indicates a clear discrepancy with the experimental data.

4) Please review all equations for typos.

Thank you, we have double-checked the equations.

Reviewer #2:[…] All of that being said, there are certain aspects of the manuscript that are rather confusing and need to be improved. Although the comments below may seem excessive to the authors, it did in fact take multiple readings and considerable additional thinking for this reviewer to fully comprehend some of what the authors had done. It can be a difficult goal to achieve, but the authors should aim for a paper that can be readily understood, even by non-experts, after a single reading. The manuscript doesn't meet that standard at present.1) One of the most interesting aspects of the manuscript is the discussion of activity coefficients, I_CaL_ current-voltage plots, and the extraction of activation curves (subsection “In-depth revision of the L-type calcium current”). The reviewer absolutely concurs that because many formulations have been inherited from previous models, often without rigorous examination of these formulations, it is entirely appropriate to examine these assumptions and re-formulate currents if necessary. However, this discussion is ultimately unconvincing, and the presentation needs to be modified for this section to be fully understood.

Thank you for highlighting the importance of this point. Following the reviewer’s suggestions, we have substantially expanded the description of the implications of the reformulation of the I_CaL_ properties, with new results and discussion. This is explained in more detailed in the following sub-points.

1a) The discussion of the activity coefficients is interesting and will be informative to readers unfamiliar with this primary literature. However, although the authors make a compelling case that different activity coefficients, i.e. closer to 0.6, should be used, the manuscript leaves out the most important question, namely: how do these changes affect the I-V curves?

Thanks for the comment. We have now introduced/expanded sections in Appendix 1, subsections “I_CaL_ updates and the I-V relationship”, “I_CaL_ reversal and ionic activity coefficients” and “Role of I_CaL_ properties and AP morphology in calcium transient changes by sodium blockade” to explain the effect of the revision of the activity coefficients more clearly. The shape of the I-V relationship is affected predominantly by the update of the activation curve as shown in Appendix 1—figure 2. This supports the case that the activation curve needs to be extracted by considering the Goldman-Hodgkin-Katz (GHK) driving force in models which use the GHK equation to compute the L-type calcium current (as explained in the Materials and methods subsection “Activation curve extraction”).

The activity coefficients themselves have an important contribution in the improved response to sodium block (Appendix 1 subsection “Role of I_CaL_ properties and AP morphology in calcium transient changes by sodium blockade”) and also in avoiding the reversal of the I_CaL_ to outward current (Figure 2F), which is exhibited by the ORd model, (Appendix 1—figure 6).

1b) Speaking of I-V curves, examination of Figure 1D makes a convincing case that the new model has a different shape and matches the experimental data better. However, it's not clear why this occurs – what allows this better fit? Is this due to the changes in activity coefficients? Is it just from the better activation curve shown in Figure 1C? Another fact that can affect this curve is the relative permeability of the channel to Na and K compared with Ca. It's odd that this is never addressed in the manuscript. Did the authors modify this to try to fit the curves better, or were these numbers considered fixed based on original data?

As mentioned in the previous comment, we have now dissected the role of activation curve versus activity coefficients in Appendix 1, subsections “I_CaL_ updates and the I-V relationship”, “I_CaL_ reversal and ionic activity coefficients” and “Role of I_CaL_ properties and AP morphology in calcium transient changes by sodium blockade”. The relative permeability to Na and K was kept as in the original ORd model; their ratio to Ca permeability was kept constant, given that the data were not challenged by new evidence.

1c) The term "driving force" is used in the manuscript in a way that is inconsistent with prior literature. One of the appeals of Ohmic formulations is that driving force is expressed in units of volts, and conductance can be derived from current plots by simple division. But with a GHK formulation, things are not so simple. The term called "driving force" here does not appear to be in units of volts, so what exactly is this term representing?

This is addressed together with the following comment.

1d) Finally, on this same topic, the Discussion states: "Activation curve of the current in previous cardiac models was based on the use of Nernst driving force in experimental studies, but the models then used Goldman-Hodgkin-Katz driving force to compute the current." Again, it's not completely clear how the authors derive one set of numbers (i.e. activation at defined voltages) from a different set of numbers (i.e. current at those same voltages). As I have been writing this, it has occurred to me that perhaps the units of driving force are less relevant, as long as one set of numbers can be multiplied by a different set of numbers to ultimately produce current. However, the fact that I've had to think so hard about this demonstrates that the manuscript needs to do better. Because the I_CaL_ formulation is claimed in the Discussion as "the greatest theoretical contribution of this work," the explanations here need to be crystal clear. Perhaps the supplement should contain flow charts of the conventional method for extracting I_CaL_ activation versus how the new approach improves this, as well as plots showing how the changes to the activity coefficients influence the IV curves.

Thank you for this suggestion. We have addressed this issue by adding a block of text at the start of the Materials and methods subsection “In-depth revision of the L-type calcium current”which clarifies the term “driving force” in the manuscript (a number that multiplies conductance/permeability and gating variables to produce total current). In addition, we have included a diagram and explanation in Appendix 1—figure 3 illustrating the extraction of activation curve. Together with the newly added Appendix subsections “ICaL updates and the I-V relationship”, “ICaL reversal and ionic activity coefficients” and “Role of ICaL properties and AP morphology in calcium transient changes by sodium blockade”, the manuscript now gives quantitative insight into the effects of the updates to I_CaL_ introduced in our work.

2) The discussion of Na current blockade and inotropy is relatively weak. Perhaps the authors need to reproduce some original experimental data, or, at the very least, provide some numbers extracted from the earlier studies. The description of Figure 3 discusses the "negative inotropy" previously reported with Na channel blockers, but with no numbers provided. The change in Ca transient amplitude shown in Figure 3C is very small, probably undetectable experimentally. If experiments have reported a 50% decrease in contraction strength, I would say that neither model is consistent, although ToR-ORd is slightly better. Obviously the heatmaps in 3E and 3F provide additional information, but in general here the correspondence with data can be discussed in a more quantitative way. In the Discussion, the phrase "good response of the ToR-ORd model to sodium blockade" sticks out as rather vague.

We have now added a more detailed comparison to available clinical data in the manuscript, subsection “Calibration: Inotropic effects of sodium blockers”).

The 15% reduction in CaT amplitude observed with sodium block in fibre simulations with the ToR-ORd model (Appendix 1 subsection “Summary of literature on sodium blockers, I_Na_, and I_NaL_ reduction”) is consistent with experimental measurements (showing changes of 9-28%, depending on the study and index of contractility). A quantitative comparison is challenging given the different measures of inotropic properties. Furthermore, the degree of I_Na_ and I_NaL_ block is estimated rather than known with precision, as also discussed in Appendix 1. Thus, the drugs studied may block I_Na_ and I_NaL_ in a different ratio than 50:50 (suggesting I_NaL_ block may be stronger than I_Na_ block; Appendix 1, subsection “Role of I_CaL_ properties and AP morphology in calcium transient changes by sodium blockade”), potentially also altering the CaT amplitude reduction in simulations.

2a) Again on the Na channel block-inotropy issue. The authors have an opportunity to obtain new mechanistic insight here given that the response of the two models is different. My suspicion is that it might have less to do with the AP morphology, as speculated in the Discussion, and more to do with intracellular [Na] regulation. However, since mechanisms are not explored, we don't know which idea is correct. It might be considered beyond the scope of this study to obtain this mechanistic insight, but it's an interesting question, and the new model provides a means for addressing it.

We have now offered mechanistic insight into this issue in Appendix 1, subsection “Calibration: I_CaL_ block and I_Kr_ replacement”. We provide the results of additional simulations using AP clamping and selective addition of I_CaL_ updates to dissect how the different updates contribute to the sodium block simulation results. The data demonstrate the importance of AP morphology, as shown in Appendix—table 1, and also Appendix 1—figure 7. We agree with the reviewer that further investigations into sodium regulation could be the focus of future studies.

3) The description of hyperkalemia simulation in Figure 8 should be improved. First, the text claims that Figure 8A shows a "progressive" increase in resting potential and slowing of upstroke velocity. But since the figure only shows 2 examples, it's hard to see that this is progressive. Second, PRR is used in this description but has not been previously defined. It took a few seconds to figure out what the authors meant. Is it really necessary to have an abbreviation for a term that's only used once in the manuscript?

Thank you for bringing this to our attention; we rewrote PRR as postrepolarization refractoriness, and we removed the “progressive”.

4) The last paragraph of the Discussion, discussing future directions, should be modified. This paragraph states: "Similarly to most existing cardiac models, the equations governing the release depend directly on the L-type calcium current, rather than on the calcium concentration adjacent to the ryanodine receptors, which is the case in cardiomyocytes. Future development of the ryanodine receptor model and calcium handling will extend the applicability of the model.…"The first sentence is correct, the second is not. The challenge is not to improve the representation of the ryanodine receptor. This will not fix the problems; the problems arise from the fact that Ca release in cells is controlled locally rather than globally – this is why whole cell models, that use a single variable to describe ryanodine receptor gating, need to use shortcuts such as making release directly dependent on I_CaL_. This general problem was rigorously analyzed by Stern way back in 1992 (PMID: 1330031). For more discussion, see also PMIDs: 15465866, 20346962, 21586292.It's great to discuss potential future improvements to the EC coupling part of the model, but these discussions should acknowledge that the issue is not just improving the RyR model, but that stochastic triggering of locally-controlled Ca sparks will need to be described.

Thank you for the comment. We have now rephrased this part of Discussion, acknowledging more possible future directions in line with the reviewer’s suggestions.

Reviewer #3:Comments:- Cogent, well-motivated Introduction. Weaknesses of ORd (AP plateau potential, APD adaptation and response to sodium current block) and solutions (L-type calcium, excitation contraction coupling and hERG current re-assessment and reformulations) clear.- Overall goal to design, develop, calibrate and validate the novel ToR-ORd model, with aim of reproducing all key depolarization, repolarization and calcium dynamics properties in healthy human ventricular cells and when these are under drug block, as well as in diseased conditions e.g. hyperkalemia and hypertrophic cardiomyopathy is both timely and useful.- Importantly, calibration and validation processes are independent, using independent datasets- "The ICaL current was deeply revisited, particularly with respect to its driving force, based on biophysical principles." – well-motivated, needed, seldom done, valuable to other models. "This allows accurate representation of the driving force when ionic concentrations are disturbed,"- Really encouraging to see disease-based, multi-scale validation of this nature. Impact for translational work made abundantly clear.Concerns/areas to address:1) Given the journal's wide appeal, there is a need to better introduce the strategy and process going into the Materials and methods section, from "We initially performed the evaluation of the ORd model (O'Hara et al., 2011) against calibration criteria…". While I appreciated the simplicity of explanation in general, this guiding summary can be improved to clarify what was initially done and why and how the strategy developed.

Thank you for this comment. We have now provided further clarifications in the section.

2) It's also not immediately clear what the calibration criteria actually are (Table 1) – I assume that, in addition to the listed references, that these exist in a supplement somewhere and it would be great to point the reader to this.

We have added a new section Appendix 1 subsection “Calibration criteria” which concretely discusses the calibration criteria in greater detail.

3) "Simulations with the existing versions of the ORd model failed to fulfil key criteria such as AP morphology, calcium transient duration, several properties of the L-type calcium current, negative inotropic effect of sodium blockers, or the depolarising effect of I_K1_ block" – where is this shown? Is there a supplementary figure?

We have now clarified where the results are shown:

“Simulations with the existing versions of the ORd model failed to fulfil key criteria such as AP morphology, calcium transient duration, several properties of the L-type calcium current, negative inotropic effect of sodium blockers, or the depolarising effect of I_K1_ block” (later shown in Figures 2, 3, and Materials and methods subsection “Calibration of I_K1_ block and resting membrane potential”).

4) In-depth revisions based on fundamental physical principles is a much-needed process. Some motivation and/or discussion of the state of the art in this particular field (i.e. how some attention to the underlying biophysics was laid by the wayside in model redevelopment, experimental assimilation, and subsequent versioning, and conjectures as to why e.g. computational tools like versioning tools did not exist, would be appropriate.

Our approach of clear calibration criteria and using the genetic algorithm for parameter fitting was very helpful in identifying properties that needed revising from fundamental physical principles. This allowed to effectively identify first the I_CaL_ properties and then the I_Kr_ structure as the two critical aspects that required in depth revisiting, based also on AP morphology. It was very useful to be able to test the effect of different I_Kr_ model structures, which needed manual coding from the papers. Initiatives such as cellML make this process more efficient, and this is why we share our code in such format. Also to facilitate versioning of this model, we are posting on github. We have now specified these points both in Materials and methods and Discussion.

5) Given the progressive, savvy lean of this work, it seems reasonable to ask why authors did not consider making the code available in other formats, e.g. python as well, or at least justify/mention the relevance of language and tools for reuse/reproducibility/versioning. Similarly, reference to other recent forays into systematic model improvement, e.g. functional curation and web lab, seems warranted.

Matlab was chosen as it’s one of the most widely used tools for cardiac simulations. Our practical experience is that it is very convenient tool for model development, given the way in which it integrates capabilities of simulation, visualisation, analysis, and introspection (via debugger which is an excellent tool to get insight into why the model behaves in a particular way).

CellML was chosen as the leading standard of model formulations, and as it can be used in simulators such as Chaste or OpenCOR, and as it may be imported into other platforms including the Myokit library which can convert CellML to Python or C, making it a very flexible format. Furthermore, we also provide an evaluation pipeline to facilitate model validation and future work. We extended the Materials and methods subsection “Evaluation pipeline and code” to explicitly point the reader to all these options, which we trust will facilitate the uptake of the model.

6) Results: "This [ToR-ORd model yields negative I_CaL_ values in such conditions] is a direct consequence of the updates to the extracellular/intracellular calcium activity coefficients, which supports its credibility and it is important for cases of elevated I_CaL_, such as under ß-adrenergic stimulation." – while this is likely true, it also seems likely a throwaway – were any simulations run using the model to demonstrate the importance in this vein? Important how?

Many thanks for raising this important question. We have now added Appendix 1, subsection “I_CaL_ reversal and ionic activity coefficients”, explaining this in greater detail, demonstrating the difference the updated activity coefficients make in the GHK equation.

7) Results/Figure 3: Besides noting that the ToR-ORd is consistent with observed negative inotropy of sodium blockers, it is unclear from text how updated model compares in terms of available experimental data, i.e. "A mild increase in inotropy may be achieved only under near-exclusive I_Na_ block." – or whether this comparison is available at all.

We agree this is very important and we’ve added a paragraph at the end of Results subsection “Calibration: Inotropic effects of sodium blockers”.

8) Figure 5A, B, C – the slope of the APD90 dependence on BCL for the ToR-ORd differs consistently from that of the data for BCLs > 1s – any musings as to mechanism? Either briefly in main text or supplement.

Our results show that the simulated data are overall within the experimental range. There is indeed a slight difference in the slope of the APD90 dependence for long BCL between the simulated data and the mean experimental recordings. The mean experimental data for each BCL are an average of many traces, some of which may be consistent with the simulation shown. In a population-of-models approach, it is likely that there will be, in turn, models highly consistent with the data means.

9) Figure 6C: would strengthen the case for the ToR-ORd further to compare restitution for ORd in a fibre under same protocol?

Thank you. We have added this to the figure.

10) Seems like suitability of ToR-ORd for drug safety testing is clear – and discussion around this is adequate and encouraging. However, there still does not seem to be a marked improvement in the example given over the ORd (Figure 7), particularly in terms of false negatives. Authors note "further improvement to the score is likely with development of a population optimised towards drug safety assessment" – so it will be interesting to see results there.

Thank you. The aim of this section of the Results was to show that the accuracy for drug-induced pro-arrhythmic risk was also high and similar to the one with the ORd. We have deleted this sentence to avoid confusion as further development to improve the accuracy, which is already high, would be beyond the scope of this work.